# ERK and USP5 govern PD-1 homeostasis via deubiquitination to modulate tumor immunotherapy

Xiangling Xiao [1,2,10], Jie Shi[1,2,10], Chuan He[1,2,10], Xia Bu[3], Yishuang Sun[1,2], Minling Gao[1,2], Bolin Xiang[1,2], Wenjun Xiong[1,2], Panpan Dai[1], Qi Mao[1,2], Xixin Xing[1,2], Yingmeng Yao[1,2], Haisheng Yu[1,2], Gaoshan Xu[1,2], Siqi Li[4], Yan Ren[5], Baoxiang Chen[6], Congqing Jiang[6], Geng Meng [7], Yu-Ru Lee [8], Wenyi Wei [9], Gordon J. Freeman [3], Conghua Xie [1] ✉ & Jinfang Zhang [1,2] ✉

The programmed cell death protein 1 (PD-1) is an inhibitory receptor on T cells and plays an important role in promoting cancer immune evasion. While ubiquitin E3 ligases regulating PD-1 stability have been reported, deubiquitinases governing PD-1 homeostasis to modulate tumor immunotherapy remain unknown. Here, we identify the ubiquitin-specific protease 5 (USP5) as a bona fide deubiquitinase for PD-1. Mechanistically, USP5 interacts with PD-1, leading to deubiquitination and stabilization of PD-1. Moreover, extracellular signal-regulated kinase (ERK) phosphorylates PD-1 at Thr234 and promotes PD-1 interaction with USP5. Conditional knockout of *Usp5* in T cells increases the production of effector cytokines and retards tumor growth in mice. USP5 inhibition in combination with Trametinib or anti-CTLA-4 has an additive effect on suppressing tumor growth in mice. Together, this study describes a molecular mechanism of ERK/USP5-mediated regulation of PD-1 and identifies potential combinatorial therapeutic strategies for enhancing anti-tumor efficacy.

Immune checkpoint inhibitors targeting PD-1/PD-L1 or CTLA-4, have been widely used for treating various types of cancer patients[1–3]. However, the moderate objective response rate to immune checkpoint blockade highlights the importance of exploring the underlying resistance mechanisms and developing improved therapeutic strategies for cancer treatment[4,5]. PD-1 (also known as CD279) is expressed on activated T cells and engages with either of its two ligands (PD-L1, also known as CD274/B7-H1 or PD-L2, also known as CD273/B7-DC) on tumor cells or other immune cells in tumor microenvironment (TME), leading to T cell dysfunction and tumor immune evasion[6–8]. Recently, investigations of PD-1/PD-L1 regulation in cancer immunotherapy have identified potential intervention targets and treatment regimens to enhance clinical efficacy[9,10]. Multiple levels of PD-L1 regulation have been extensively studied[11], while

[1]Department of Radiation and Medical Oncology, Hubei Key Laboratory of Tumor Biological Behaviors, Hubei Cancer Clinical Study Center, Zhongnan Hospital of Wuhan University; Medical Research Institute, Frontier Science Center of Immunology and Metabolism, Wuhan University, Wuhan 430071, China. [2]Taikang Center for Life and Medical Sciences, Wuhan University, Wuhan 430071, China. [3]Department of Medical Oncology, Dana-Farber Cancer Institute, Harvard Medical School, Boston, MA 02115, USA. [4]Department of Biology, University of Copenhagen, Copenhagen 2100, Denmark. [5]Experiment Center for Science and Technology, Shanghai University of Traditional Chinese Medicine, Shanghai 201203, China. [6]Department of Colorectal and Anal Surgery, Low Rectal Cancer Diagnosis and Treatment Center, Zhongnan Hospital of Wuhan University, Wuhan 430071, China. [7]College of Veterinary Medicine, China Agricultural University, Beijing 100094, China. [8]Institute of Biomedical Sciences, Academia Sinica, Taipei 115201, Taiwan. [9]Department of Pathology, Beth Israel Deaconess Medical Center, Harvard Medical School, Boston, MA 02115, USA. [10]These authors contributed equally: Xiangling Xiao, Jie Shi, Chuan He. ✉e-mail: chxie_65@whu.edu.cn; jinfang_zhang@whu.edu.cn

investigations of PD-1 regulation, especially at the post-translational level, are relatively limited.

The N-linked glycosylation is a post-translational modification (PTM), which is crucial for membrane protein folding, stability, and localization[12,13]. Recent studies suggest that PD-1 can be modified with the N-linked glycosylation that maintains PD-1 protein stability, localization, and its binding with approved PD-1-specific therapeutic antibodies[14,15]. Moreover, the N-glycans of PD-1 are highly core fucosylated, leading to the stabilization of PD-1 largely through preventing the ubiquitin E3 ligase FBXO38-mediated ubiquitination and degradation of PD-1[16,17].

Ubiquitination is another important type of PTM and plays a major role in regulating protein stability and interaction to maintain various cellular processes[18,19]. Ubiquitination and deubiquitination are reversible processes, which are governed by the E3 ligases and deubiquitinating enzymes (DUBs) to control protein homeostasis in cells[19,20]. Several ubiquitin E3 ligases including FBXO38, KLHL22 and c-Cbl have been reported to promote PD-1 ubiquitination and subsequent degradation[21–23], leading to the enhanced anti-tumor response of T cells. However, the physiological deubiquitinase that regulates PD-1 stability remains unknown.

Here, we show that the ubiquitin-specific protease 5 (USP5) functions as a bona fide deubiquitinase for PD-1. Moreover, the extracellular signal-regulated kinase (ERK) directly phosphorylates PD-1 at Thr234 residue in the cytoplasmic tail to stabilize PD-1 largely through promoting the PD-1/USP5 interaction, leading to reduced ubiquitination and degradation of PD-1. Furthermore, ablation of *Usp5* in CD8[+] T cells decreases PD-1 protein expression and increases the production of cytokine and cytotoxic molecules, including interferon-gamma (IFN-γ), tumor necrosis factor (TNF), and granzyme B (GzmB) upon stimulation with anti-CD3/CD28 antibodies in vitro. Mice with conditional knockout (cKO) *Usp5* in T cells have better tumor control than wild-type (WT) mice. Importantly, USP5 inhibition in combination with the MEK inhibitor, Trametinib, or anti-CTLA-4 antibodies has an additive effect on suppressing tumor growth in mice.

## Results

### Identifying the deubiquitinase USP5 as a positive regulator for PD-1

Ubiquitination/deubiquitination is a reversible post-translational process governed by E3 ligases and deubiquitinases, which dynamically and tightly regulate protein homeostasis to control various cellular processes[19,20]. To identify deubiquitinases for regulating PD-1 stability, we performed the immunoprecipitation using anti-HA antibody from cell lysates of HEK293T cells transfected with PD-1-cHA or empty vector (EV) as a negative control, and the immuno-complex was analyzed by mass spectrometry (MS) to identify potential PD-1-interacting proteins in cells (Fig. 1a, Supplementary Fig. 1a, and Supplementary Data 1). Previously reported PD-1-interacting proteins, the ubiquitin E3 ligase adaptor protein KLHL22[22], the scaffold protein Cullin 3, together with KLHL22 and RBX1 forming a functional E3 ligase complex, and the tyrosine phosphatase SHP2 encoded by the *PTPN11* gene[24,25], were identified in the anti-HA immuno-complex from cell lysates expressing PD-1-cHA, but not in the immuno-complex from cell lysates expressing EV (Fig. 1a and Supplementary Data 1). These previously reported PD-1-associated proteins confirmed that our screen for identifying PD-1-interacting proteins was valid.

Of note, four deubiquitinases including USP5, USP10, USP15, and OTUB1, were identified in the immuno-complex from cell lysates expressing PD-1, indicating that these DUBs might be the potential PD-1-interacting proteins (Fig. 1a and Supplementary Data 1). Moreover, a recent study also identified USP5, USP10, and USP15 as the potential PD-1-interacting proteins by the MS analysis, albeit they did not further study whether and how these DUBs regulate PD-1 in cells[22]. To further confirm which DUB is a bona fide deubiquitinase for PD-1, we

ectopically co-expressed these four DUBs and PD-1 to examine their effects on PD-1 protein abundance, respectively. Our results demonstrated that USP5 and USP10, but neither USP15 nor OTUB1, dramatically upregulated PD-1 protein levels in cells (Fig. 1b and Supplementary Fig. 1b–d). However, only the enzymatically inactive USP5 (USP5-C335A)[26], but not the inactive USP10 (USP10-C424A)[27], failed to elevate PD-1 protein abundance, indicating that USP5, but not USP10, stabilizes PD-1 through its deubiquitinating enzyme activity (Fig. 1b and Supplementary Fig. 1b). Ectopic expression of USP5 did not significantly alter the *PD-1* mRNA levels, suggesting that USP5-mediated upregulation of PD-1 protein abundance might be at the post-translational level (Fig. 1c). We examined the PD-1 protein expression level in different cell lines and found that Jukat and MOLT-4 cells had relatively high PD-1 expression upon phytohemagglutinin A (PHA) stimulation (Supplementary Fig. 1e). Next, we investigated whether knockdown of *USP5* decreases PD-1 protein expression in both Jukat and MOLT-4 cells. We designed and constructed shRNAs targeting the coding sequence for each gene of *USP5*, *USP10*, and *USP15* genes (Supplementary Fig. 1f). Knockdown of *USP5*, but neither *USP10* nor *USP15*, markedly decreased endogenous PD-1 protein levels in MOLT-4 and Jurkat cells (Fig. 1d and Supplementary Fig. 1g–j). Moreover, knockdown of *USP5* also did not result in significant change of *PD-1* mRNA (Supplementary Fig. 1k), further supporting the notion that USP5 regulates PD-1 protein abundance at the post-translational level. Together, these results suggest that USP5 is the major deubiquitinase for regulating PD-1 protein homeostasis in cells. Thus, we focused on exploring PD-1 regulation by USP5 in the remaining study.

To confirm whether USP5 stabilizes PD-1 at the post-translational level, we utilized the cycloheximide (CHX) to inhibit protein translation and analyzed the effect of USP5 on PD-1 protein half-life. Ectopic expression of wild-type (WT) USP5, but not enzymatically inactive mutant C335A, significantly prolonged the half-life of PD-1 protein in cells (Fig. 1e, f, and Supplementary Fig. 1l, m). Conversely, knockdown of *USP5* significantly shortened the PD-1 protein half-life in both Jurkat and MOLT-4 cells (Fig. 1g, h, and Supplementary Fig. 1n, o). These results demonstrate that USP5 positively regulates PD-1 protein stability primarily at the post-translational level.

As DUBs play critical roles in tumorigenesis, small-molecule inhibitors targeting the enzymatic activities of DUBs have been designed and synthesized for cancer treatment in preclinical studies or clinical trials[28]. We applied a panel of DUBs inhibitors to treat cells and found that the EOAI3402143 was the most potent one to dramatically decrease PD-1 protein abundance in MOLT-4, mouse primary CD3[+] T cells, and Jurkat cells (Fig. 1i, j, and Supplementary Fig. 1p). Moreover, PD-1 protein expression, but not its mRNA level, was gradually reduced following increased concentration of EOAI3402143 treatment (Fig. 1k–p, and Supplementary Fig. 1q). EOAI3402143 treatment dramatically reduced the protein abundance of PD-1, but not other immune checkpoints, including PD-L1 and VISTA, indicating that EOAI3402143 might specifically regulate PD-1 in cells (Supplementary Fig. 1r). EOAI3402143 has been identified as a general inhibitor that simultaneously targets USP5, USP9x, and USP24[29,30]. However, knockdown of USP9x or USP24 using shRNAs did not obviously affect PD-1 protein abundance in Jurkat cells, suggesting that EOAI3402143 treatment reduces PD-1 protein abundance largely through targeting the deubiquitinase USP5 (Supplementary Fig. 1s, t). To further explore the clinical significance, we examined the protein expression of USP5 and PD-1 in tumor tissues of human colorectal cancer patients by using the multiplexed immunohistochemistry (mIHC) staining assay. Our results showed that USP5 colocalized with PD-1 on tumor-infiltrating CD3[+] T cells in human colorectal cancer samples (Fig. 1q). Previous studies demonstrated that although USP5 has a cytoplasmic distribution, its major localization is in the nucleus, where USP5 is involved in regulating the DNA damage repair or heat-induced stress granules in cancer cells[31,32]. However, a more recent study showed that USP5

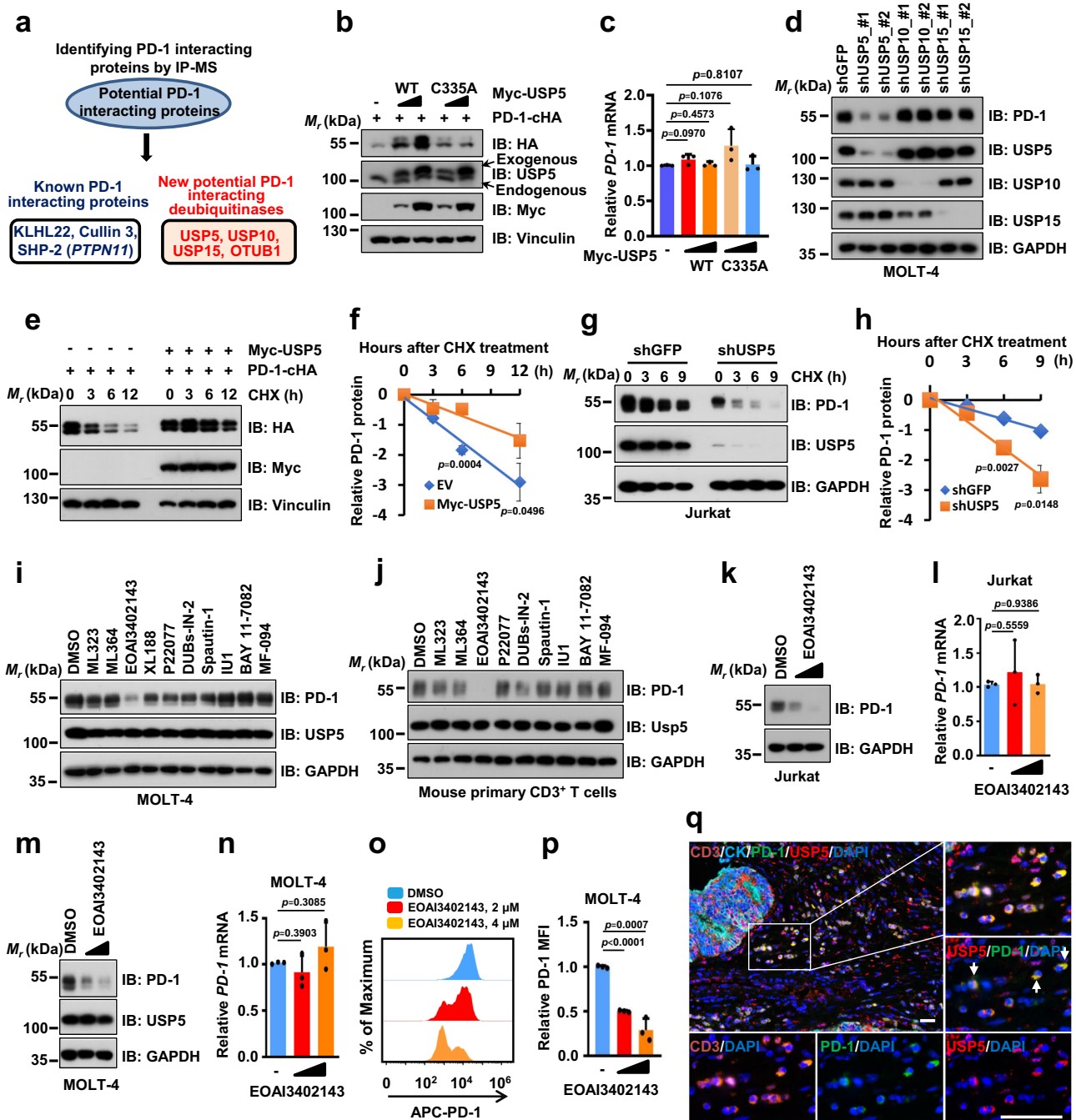

**Fig. 1 | Identifying the deubiquitinase USP5 as a positive regulator for PD-1.**
**a** Immunoprecipitation (IP) coupled with mass spectrometry (MS) analysis to identify potential PD-1-interacting protein including indicated deubiquitinases. Potential PD-1-interacting proteins purified from HEK293T cells transfected with pcDNA3.1-PD-1-cHA were analyzed by MS. All proteins were identified in Supplementary Data 1. **b**–**d** Immunoblotting (IB) analysis of whole-cell lysates (WCL) from HEK293T cells co-transfected with indicated constructs (**b**) or MOLT-4 cells stably expressing shUSP5, shUSP10, shUSP15 or shGFP (**d**). The mRNA level of PD-1 was measured using quantitative reverse transcription PCR (qRT-PCR) (**c**). **e**–**h** IB analysis of WCL from HEK293T cells transfected with indicated constructs (**e**) or Jurkat cells stably expressing shUSP5 as well as shGFP (**g**). Cells were treated with 200 μg/mL cycloheximide (CHX) for indicated time points. PD-1 band intensity was quantified by Image J, which was normalized to Vinculin/GAPDH and then compared to the t = 0 time point (**f**, **h**). **i**, **j** IB analysis of WCL from MOLT-4 cells (**i**) or mouse primary CD3+ T cells (**j**) treated with indicated deubiquitinase inhibitors (1.5 μM) for

8 h. Mouse primary CD3+ T cells were pre-stimulated with anti-CD3/CD28 (2 μg/mL) for 48 h. **k**–**n** IB analysis of WCL from Jukat or MOLT-4 cells treated with EOAI3402143 (1 or 1.5 μM) for 8 h (**k**, **m**). The mRNA level of PD-1 was measured using qRT-PCR (**l**, **n**). **o**, **p** Cell surface PD-1(**o**) and relative mean fluorescence intensity (MFI) of PD-1 (**p**) on MOLT-4 cells was analyzed by flow cytometry.
**q** Representative multiplex immunohistochemistry (mIHC) images of USP5 (red), PD-1 (green), CD3 (Orange), CK (cyan), and DAPI nuclear staining (blue) in human colon tumor sections. White arrows indicate positive cells for PD-1 and USP5 colocalization. Yellow color is considered overlapping for PD-1 and USP5 staining. n = 5. Scale bars, 50 μm. For **g**, **k**, Jurkat cells were stimulated with PHA (150 ng/mL) for 3 days. For **c**, **f**, **h**, **l**, **n**, and **p**, data were presented as mean ± S.D. n = 3 biologically independent samples. Two-tailed unpaired t-test. All IB data are representative of three independent experiments. Source data are provided as a Source data file.

largely localizes in the cytoplasm, whether USP5 interacts with NLRP3 to govern inflammasome activation[33]. Our results showed that USP5 largely localizes in the cytoplasm of tumor-infiltrating CD3[+] T cells albeit some signal for USP5 can be observed in the nuclear (Fig. 1q). Together, these studies suggest that the predominant localization of USP5 in the cytoplasm or nucleus might be related to the cell types or different pathological conditions. Together, these results indicate that high expression of USP5 in T cells might stabilize PD-1 to suppress the cytotoxic function of T cells, leading to promoting tumorigenesis.

## USP5 interacts with PD-1 and deubiquitinates PD-1

As USP5 functions as a deubiquitinase and moves ubiquitination on its target protein, we hypothesized that USP5 might interact with PD-1 and remove ubiquitination on PD-1 to prevent its degradation, leading to PD-1 stabilization. In keeping with this hypothesis, our co-immunoprecipitation (co-IP) assay showed that USP5, but not USP10, USP15, nor OTUB1, had a strong binding affinity with PD-1 in cells (Supplementary Fig. 2a). The interaction between USP5 and PD-1 at endogenous level was detected in Jurkat, MOLT-4, and mouse primary CD3[+] T cells (Fig. 2a, b, and Supplementary Fig. 2b). Moreover, the GST pull-down assay demonstrated that bacterially purified recombinant GST-USP5, but not GST protein, interacted with PD-1 (Fig. 2c). It has been reported that USP5 contains an N-terminal cryptic ZnF domain, a ZnF-UBP domain, a ubiquitin-specific protease (USP) domain, and two ubiquitin-binding-associated domains (UBA1 and UBA2)[34]. To examine which domain(s) on USP5 was important for interaction with PD-1, we generated various USP5 truncations lacking different domains as indicated (Fig. 2d). Our results demonstrated that the USP5-ΔUBA1 or ΔUBA2 exhibited a lower binding affinity with PD-1 comparing with USP5-WT or ΔZnF-UBP (Supplementary Fig. 2c). However, the USP5-ΔUBA1/2 mutant lacking both UBA1 and UBA2 domains totally lost its capability to interact with PD-1 (Fig. 2e, f), suggesting that USP5 interacts with PD-1 through both UBA1 and UBA2 domains.

As the PD-1 protein contains an extracellular domain, a transmembrane domain, and a cytoplasmic region[35], we truncated PD-1 from its cytoplasmic tail and generated the PD-1-ΔC241-288 (deleting amino acid residues from 241 to 288) and ΔC192-288 (missing all cytoplasmic region) mutants, respectively (Fig. 2g). While the PD-1-ΔC241-288 maintained the capability to interact with USP5, the PD-1-ΔC192-288 failed to interact with USP5 (Fig. 2h and Supplementary Fig. 2d), suggesting the region of amino acid residues from 192 to 240 in the cytoplasmic part of PD-1 might be critical for interaction with USP5.

Furthermore, the in vivo ubiquitination assay showed that ectopic expression of USP5, but not USP10, specifically removed the ubiquitination on PD-1 in cells (Fig. 2i). Moreover, the enzymatic inactive form of USP5-C335A lost its capability to deubiquitinate PD-1 (Fig. 2j), suggesting that USP5 removes the ubiquitination on PD-1 through its deubiquitinating enzymatic activity. Previous reports demonstrated that PD-1 could be modified with K48-linked ubiquitination, leading to its degradation[21,22]. We utilized the ubiquitin K48-only construct to keep only the K48 residue and change all the remaining six lysine residues to arginine on the ubiquitin molecule. Consistently, PD-1 could be heavily modified with K48-linked ubiquitination and ectopic expression of USP5 obviously removed the K48-linked ubiquitination on PD-1 (Fig. 2k). Moreover, the purified recombinant USP5 from baculovirus-insect cells or bacteria could directly deubiquitinate PD-1 in vitro (Fig. 2l, and Supplementary Fig. 2e, f). Furthermore, we also purified endogenous USP5 protein using anti-USP5 antibodies from mouse primary CD3[+] T cells after stimulation with anti-CD3/CD28 antibodies and ubiquitinated PD-1 from HEK293T cells transfected with PD-1-cHA and His-ubiquitin. Subsequently, the in vitro deubiquitination assay was performed by co-incubating purified USP5 protein with ubiquitinated PD-1 with/without EOAI3402143 treatment. The results demonstrated that purified USP5 dramatically removed the

ubiquitination on PD-1, while EQAI3402143 could inhibit the USP5-mediated deubiquitination on PD-1, providing evidence that the enzymatic activity of USP5 in removing PD-1 ubiquitination is directly inhibited in vitro (Supplementary Fig. 2g). Conversely, knockdown or knockout of *USP5* markedly elevated endogenous K48-linked ubiquitination on PD-1 in Jurkat or mouse primary CD3[+] T cells (Fig. 2m, n). These results together suggest that USP5 specifically interacts with PD-1 and removes the ubiquitination of PD-1 in cells.

## ERK stabilizes and phosphorylates PD-1 at Thr234

Phosphorylation and ubiquitination are two important post-translational modifications and their interplay plays a critical role in regulating the fate and function of a protein[36,37]. Song et al. reported that JNK1-mediated phosphorylation of NLRP3 at Ser194 is essential for NLRP3 deubiquitination and subsequent inflammasome activation[38]. Moreover, a recent study also showed that the phosphorylation of METTL3 by ERK increases the deubiquitination of METTL3, leading to the stabilization of METTL3[39]. In our mass spectrometry analysis results (Supplementary Data 1), several kinases were also on the list of PD-1-interacting protein candidates. To explore whether a kinase can affect PD-1 protein abundance, we screened a panel of kinases, not only including PGK1, HK2 and NEK9 on the list of potential PD-1-interacting proteins, but also those involving several important kinases in cells, such as ERK1, GSK3β, CDKs and AKT (Fig. 3a). Among these kinases we examined, ERK1 was the most potent kinase to elevate PD-1 protein abundance in cells (Fig. 3a, b, and Supplementary Fig. 3a). Moreover, ectopic expression of the ERK upstream activators[40], RAS or BRAF, also elevated the PD-1 protein expression levels in cells (Fig. 3c and Supplementary Fig. 3b). Consistently, activating the ERK signaling in lung tumor tissues of *Kras*[LSL-G12D/+]*Trp53*[fl/fl] (KP) mice induced by intranasal instillation of Adenovirus-Cre (Ad-Cre) also elevated PD-1 protein abundance in CD3[+] T cells derived from lung tumor tissues, but not normal lung tissues (Supplementary Fig. 3c). Furthermore, the mIHC staining assay showed that the phosphorylated ERK (p-ERK), indicator for ERK signaling activation, had a colocalization with PD-1 in tumor-infiltrating CD3[+] T cells in human colorectal cancer samples (Fig. 3d). In contrast, suppressing the ERK kinase by knockdown of *ERK1/2* expression or pharmacological inhibitors reduced PD-1 protein abundance in Jurkat cells (Fig. 3e–g and Supplementary Fig. 3d–h). Moreover, ectopic expression of ERK1 dramatically extended the half-life of PD-1 protein (Supplementary Fig. 3i), indicating that the ERK signaling might stabilize PD-1 at the post-translational level.

Through examining the PD-1 protein sequence, we found that there is a potential conserved ERK binding D motif in PD-1 cytoplasmic region $((K/R)_{0-2}\text{-}(X)_{1-6}\text{-}\Phi\text{-}X\text{-}\Phi$, X is any amino acid and Φ is a hydrophobic residue)[40], suggesting that PD-1 might interact with the ERK kinase through its D motif (Fig. 3h). In keeping with this notion, PD-1-K210E with a mutation of the lysine (K) into glutamic acid (E) in D domain disrupted its interaction with ERK1/2 (Fig. 3i, j, and Supplementary Fig. 3j, k). Since ERK kinase interacted with PD-1, we wondered whether ERK could directly phosphorylate PD-1. The in vitro kinase assay showed that compared with the PD-1 extracellular domain (aa23-170), the cytoplasmic region of PD-1 (aa192-288) was heavily phosphorylated by ERK1 kinase (Fig. 3k). One potential ERK substrate consensus motif (serine/threonine-proline, S/T-P: $T_{234}P_{235}$ for human PD-1; $T_{236}P_{237}$ for mouse PD-1) exists in the cytoplasmic region of PD-1, indicating that Thr234 on PD-1 might be an ERK phosphorylation site (Supplementary Fig. 4a). Consistently, mutating Thr234 to Ala (T234A) largely eliminated ERK-mediated phosphorylation of PD-1 in vitro (Fig. 3l). Furthermore, the result of mass spectrometry analysis demonstrated that the T234 of PD-1 was phosphorylated (Supplementary Fig. 4b), further confirming our finding that T234 on PD-1 is an ERK phosphorylation site in cells.

We further developed the phosphorylated-T234-PD-1 (pT234-PD-1) antibodies and proved that the generated antibodies specifically

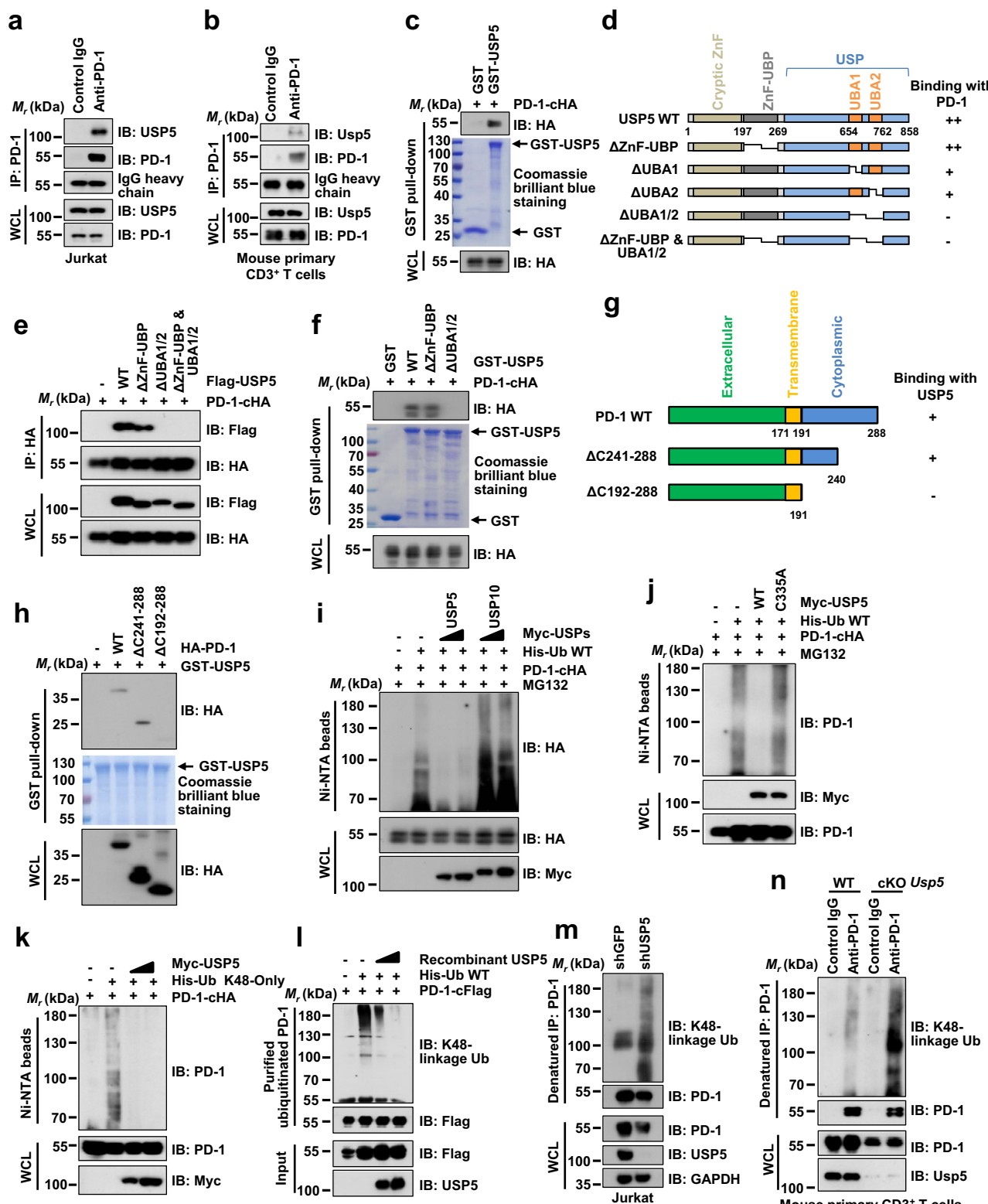

recognized phosphorylated-T234, but not unphosphorylated-T234-PD-1 synthetic peptides (Supplementary Fig. 4c). Moreover, the pT234-PD-1 antibody could recognize PD-1 WT, but not PD-1 T234A mutant in cells (Supplementary Fig. 4d). Of note, pT234-PD-1 was markedly induced by ectopic expression of ERK or BRAF in cells (Fig. 3m, n). However, the PD-1 K210E mutant that failed to interact with ERK was not detected by pT234-PD-1 specific antibodies even upon ERK activation (Supplementary Fig. 4e, f). Furthermore, the phosphorylation of PD-1 at ThrT234 was detected at the endogenous level and suppressing

ERK activity by Trametinib or Ulixertinib dramatically reduced the pT234-PD-1 level in Jurkat and MOLT-4 cells (Fig. 3o–q, and Supplementary Fig. 4g–i). Of note, reduction of *Erk1/2* expression, but not the *Usp5* expression, in mouse primary CD3⁺ T cells dramatically decreased the level of PD-1 phosphorylation at T234 site (Supplementary Fig. 4j, k). These results together demonstrate that ERK directly interacts with PD-1 and phosphorylates PD-1 on Thr234.

The N-linked glycosylation, a post-translational protein modification that largely occurs in the Endoplasmic Reticulum (ER) and the

**Fig. 2 | USP5 interacts with PD-1 and deubiquitinates PD-1. a–c** IB analysis of WCL and anti-PD-1 IPs derived from Jurkat cells (**a**), mouse primary CD3⁺ T cells (**b**), or glutathione S-transferase (GST) pull-down precipitates from HEK293T cell lysates with ectopic expression of PD-1-cHA incubated with recombinant GST or GST-USP5 protein (**c**). Jurkat cells were stimulated with PHA (150 ng/mL) for 3 days. Mouse primary CD3⁺ T cells were stimulated with anti-CD3/CD28 (2 μg/mL) for 48 h. **d**, **g** A schematic illustration of USP5 (**d**) or PD-1 (**g**) protein sequence to show its different domains and deletion truncations we generated. **e**, **f**, **h** IB analysis of WCL and anti-HA IPs (**e**) or GST pull-down precipitates (**f**, **h**) from HEK293T cell lysates co-transfected with indicated constructs. **i**–**k** IB analysis of WCL and Ni-NTA pull-down products derived from lysates of HEK293T cells transfected with the indicated constructs. Cells were treated with 10 μM MG132 for 12 h before harvesting. **l** For

in vitro deubiquitination assay, HEK293T cells transfected with His-ubiquitin and PD-1-cFlag were treated with 10 μM MG132 for 12 h. Ubiquitinated PD-1 was purified with IP using anti-Flag beads and was incubated without or with purified recombinant USP5. IB analysis with indicated antibodies. **m** IB analysis of WCL and anti-PD-1 denatured-IPs derived from lysates of shGFP- or shUSP5-treated Jurkat cells using indicated antibodies. Cells were treated with PHA (150 ng/mL) for 3 days and 5 μM MG132 for 6 h. **n** IB analysis of WCL and anti-PD-1 denatured-IPs derived from naïve CD3⁺ T cells isolated from spleens of *Usp5*^fl/fl^ (WT) or *Usp5*^fl/fl^ *Cd4*-Cre (cKO) mice. CD3⁺ T cells were stimulated with anti-CD3/CD28 (2 μg/mL) for 48 h and with 5 μM MG132 for 4 h before harvesting. All data are representative of two independent experiments. Source data are provided as a Source data file.

Golgi apparatus, is crucial for membrane protein folding, stability, and localization[12,13]. The PD-1 protein is modified with the N-linked glycosylation at N49, N58, N74, and N116, which is critical for maintaining PD-1 protein stability and localization[14]. To determine whether the ERK/USP5 signaling regulates PD-1 localization in ER and Golgi apparatus, we isolated the ER or Golgi from MOLT-4 cells using the ER or Golgi enrichment kit. The results showed that ERK, USP5, and PD-1 existed in both the ER and Golgi apparatus albeit the protein level of USP5 and PD-1 was relatively low in the Golgi apparatus compared to ER (Supplementary Fig. 5a, b). Furthermore, the immunofluorescence (IF) assay demonstrated that knockdown of *ERK1/2* or *USP5* dramatically decreased the PD-1 protein expression, leading to disrupting its localization in the ER and Golgi apparatus (Supplementary Fig. 5c–g). Overall, these results suggest the ERK/USP5 signaling pathway might regulate PD-1 stability to affect its localization on ER and Golgi apparatus in T cells.

To further explore whether the N-linked glycosylation affects the ERK-mediated phosphorylation of PD-1 at T234, we first generated the PD-1 4NQ (N49Q/N58Q/N74Q/N116Q) mutant with substitution of all four sites asparagine (N) with glutamine (Q) and found that the 4NQ mutant reduced the molecular weight of the PD-1, suggesting that PD-1 lost its N-linked glycosylation (Supplementary Fig. 5h), which is consistent with the previous report[14]. The signal for T234 phosphorylation was still detected on the PD-1 4NQ mutant with the anti-pT234-PD-1 antibody, indicating that missing the N-linked glycosylation on PD-1 does not affect the phosphorylation of PD-1 at the T234 site (Supplementary Fig. 5h). However, although the PD-1 T234A mutant lost its phosphorylation, the glycosylation on PD-1 T234A mutant could be detected comparably as the wild-type PD-1, suggesting that the phosphorylation deficiency at T234 on PD-1 may not affect its glycosylation (Supplementary Fig. 5h). In addition, treatment with the inhibitor of N-linked glycosylation, tunicamycin, reduced the N-linked glycosylation on both PD-1 WT and T234A mutant (Supplementary Fig. 5i). However, the tunicamycin-induced un-glycosylation form of PD-1 WT, but not the T234A mutant, could be phosphorylated at T234 site (Supplementary Fig. 5i). Taken together, these results demonstrate that the N-linked glycosylation and phosphorylation at T234 on PD-1 might not affect each other and should be two parallel pathways to regulate PD-1 stability and localization at different physiological or pathological conditions.

## ERK-mediated Thr234 phosphorylation of PD-1 promotes its interaction with USP5

We further tested whether ERK-mediated phosphorylation of PD-1 affected its interaction with USP5. In agreement with that ERK signaling activation stabilized PD-1 in cells, the co-IP assay showed that ectopic expression of ERK or its upstream activators KRAS or BRAF, dramatically enhanced the interaction between PD-1 and USP5 (Fig. 4a–c, and Supplementary Fig. 6a, b). In contrast, the co-IP and GST pull-down assays demonstrated that suppressing ERK activity by Trametinib or Ulixertinib dramatically reduced the PD-1-USP5 interaction in cells (Fig. 4d–f and Supplementary Fig. 6c, d). In keeping with this result, the

phosphorylation-deficient mutant PD-1-T234A displayed a reduced interaction with USP5 (Fig. 4g, h). Furthermore, the recombinant human USP5 protein had a high affinity with the ERK1-phosphorylated GST-PD-1 (192-288) in vitro (Fig. 4i). Furthermore, only the synthetic PD-1 peptides with T234 phosphorylation interacted with bacterially purified GST-USP5 WT and UBA1/2 domains, but not with USP5 N-terminal cryptic ZnF domain and ZnF-UBP domain in vitro (Fig. 4j, k, and Supplementary Fig. 6e). To identify the critical amino acid residues in UBA1/2 domain of USP5 that interacts with PD-1, we aligned protein sequences and compared crystal structures of UBA1 and UBA2 domains in USP5 (Supplementary Fig. 6f, g). Several conserved amino acid residues located on the surface of crystal structure of UBA1 and UBA2 domains were identified, which might be potential key amino acid residues to mediate interaction with PD-1 in cells (Supplementary Fig. 6f, g). These conserved amino acid residues were divided into four groups based on their positions on the crystal structure of UBA1 and UBA2 domains (Supplementary Fig. 6g). To identify specific residues on UBA1/2 domains for PD-1 interaction, we mutated these conserved amino acid residues in each group into alanine residues on the UBA1/2 domains of USP5, designating the mutations as M1, M2, M3, and M4, respectively (Supplementary Fig. 6g). We performed the streptavidin pull-down assays by incubating biotin-labeled synthetic pT234-PD-1 peptides with Flag-USP5 WT or different mutants. Compared with USP5 WT and other mutants, the interaction between M4 and pT234-PD-1 peptide was obviously reduced (Supplementary Fig. 6h), suggesting the specific amino acid residues in M4 on UBA1/2 domains of USP5 are critical for interaction with PD-1.

As USP5 interacted with PD-1 and removed the ubiquitination on PD-1 (Fig. 2), we investigated how the ERK-mediated phosphorylation affected the level of PD-1 ubiquitination. Compared with PD-1 WT, the phosphorylation-deficient PD-1-T234A mutant was heavily ubiquitinated and became resistant to USP5-mediated deubiquitination (Fig. 4l). Inhibiting ERK activation by Trametinib suppressed the USP5-mediated deubiquitination of PD-1 (Fig. 4m). Moreover, inhibition of ERK activity by knockdown of *ERK1/2* markedly elevated the endogenous ubiquitination of PD-1 in Jurkat cells (Fig. 4n). To address whether the mutation of a single phosphorylated site on PD-1 abrogates the effects of USP5 on modulating PD-1 stability in primary T cells, we introduced the human PD-1 WT and the T234A mutant into the mouse primary CD3⁺ T cells using the retro-virus expression system. The results showed that increased concentration of EOAI3402143 treatment gradually decreased the protein level of PD-1 WT, but not the T234A mutant, in the mouse primary CD3⁺ T cells, suggesting that the single phosphorylation site mutation on PD-1 could abolish USP5-mediated regulation of PD-1 stability (Supplementary Fig. 6i, j). Together, these results suggest that phosphorylation of PD-1 at Thr234 by ERK promotes PD-1 interaction with USP5, leading to deubiquitination and stabilization of PD-1.

## *Usp5* conditional knockout (cKO) mice have effective tumor control

To examine whether USP5 expression affects the cytotoxic activity of CD8⁺ T cells, we performed in vitro T cell killing assays. Activated OT-1

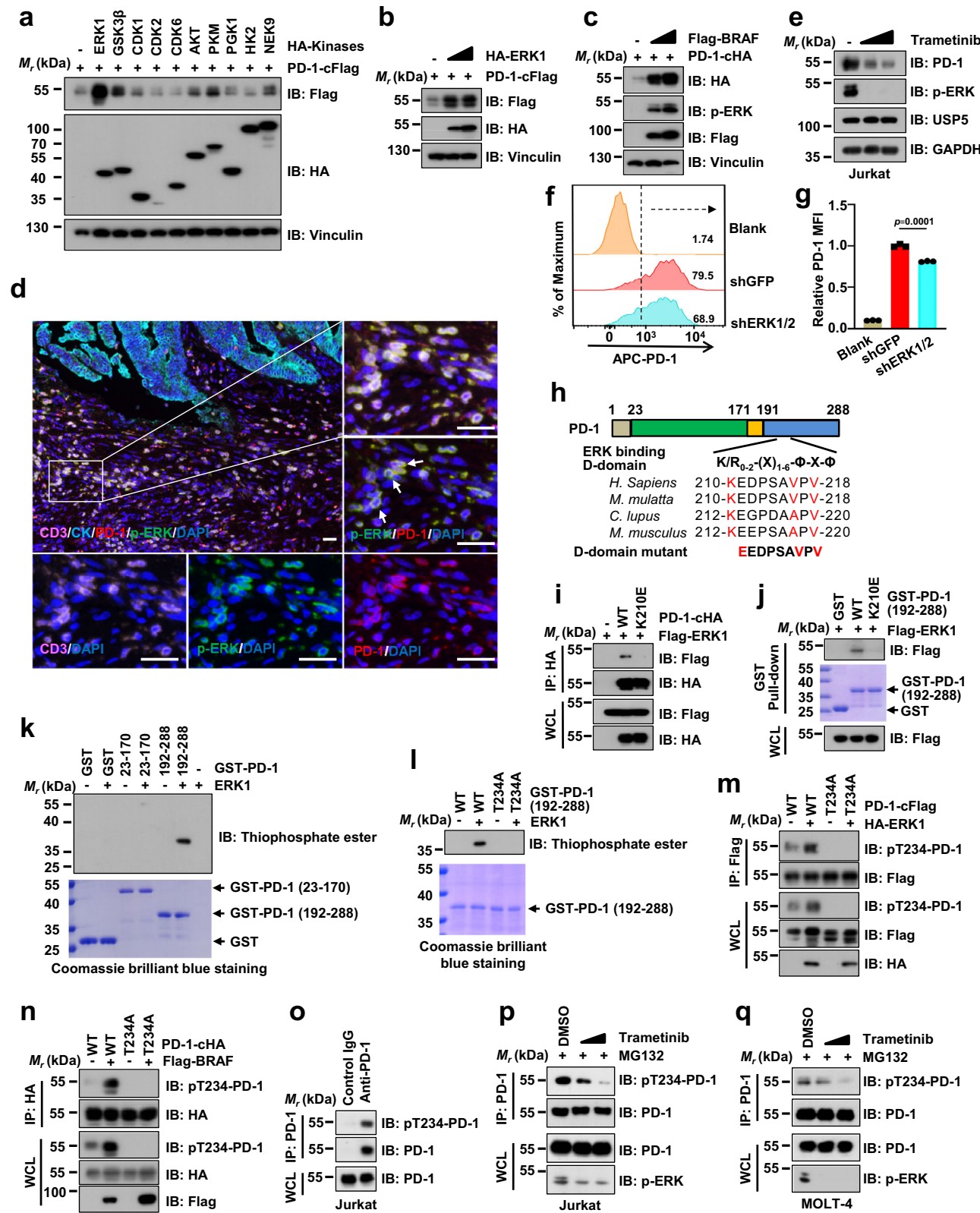

CD8[+] T cells were infected with lentiviruses expressing shUsp5 to knockdown *Usp5* expression, resulting in reduced PD-1 protein abundance (Supplementary Fig. 7a). Apoptosis of MC38 tumor cells was analyzed after co-culturing shRNA-treated OT-1 CD8[+] T effector cells (E) with OVA-pulsed MC38 tumor target cells (T)[41]. Compared with the control group, apoptosis of MC38 cells was significantly increased in the group of co-culture with *Usp5*-deficient OT-1 CD8[+] T cells

(Supplementary Fig. 7b), suggesting that knockdown of *Usp5* enhances the cytotoxic activity of CD8[+] T cells towards MC38 tumor cells in vitro.

To further explore the role of USP5 in regulating T cell function and tumor growth in vivo, *Usp5* cKO (*Usp5^flox/flox^Cd4-Cre*) mice with specifically deleting *Usp5* in T cells were generated through crossing *Usp5^flox/flox^* mice with *Cd4-Cre* mice (Fig. 5a, b). Compared with WT mice, *Usp5* cKO mice did not display obvious effects on embryonic

**Fig. 3 | ERK phosphorylates PD-1 at Thr234 to stabilize PD-1. a–c** IB analysis of WCL derived from HEK293T cells co-transfected PD-1-cFlag or PD-1-cHA with different kinases as indicated. **d** Representative mIHC images of CD3 (pink), PD-1 (red), phosphor-ERK at Thr202/Try204 (p-ERK, green), CK (cyan), and DAPI nuclear staining (blue) in human colon tumor sections. White arrows indicate positive cells for PD-1 and p-ERK colocalization. Yellow color is considered overlapping for PD-1 and p-ERK staining. *n* = 5. Scale bars, 50 μm. **e** IB analysis of WCL derived from Jurkat cells treated with PHA (150 ng/mL) for 3 days and Trametinib (1 or 3 μM) for 24 h before harvesting. **f, g** Cell surface PD-1 on shGFP- or shERK1/2-treated Jurkat cells with pre-stimulation of PHA (150 ng/mL) for 3 days was analyzed by flow cytometry (**f**). The relative mean fluorescence intensity (MFI) of PD-1 on the surface of was quantified (**g**). Data were presented as mean ± S.D. *n* = 3 biologically independent samples per group. Two-tailed unpaired *t*-test. **h** A schematic illustration and sequence alignment of a potential ERK binding D-domain on PD-1 protein sequence, $(K/R)_{0-2}$-$(X)_{1-6}$-Φ-X-Φ, where Φ is a hydrophobic residue and X is any amino acid. **i, j** IB analysis of WCL and anti-HA IPs (**i**) or GST pull-down precipitates (**j**) from HEK293T cell lysates transfected with indicated constructs. **k, l** In vitro phosphorylation assays of bacterially purified recombinant GST, GST-PD-1 truncations or T234A mutant by ERK1 kinase. **m, n** IB analysis of WCL and anti-Flag or anti-HA IPs derived from HEK293T cells transfected with indicated constructs. **o–q** IB analysis of WCL and anti-PD-1 IPs from Jurkat pre-treated with PHA (150 ng/mL) for 3 days (**o, p**) or MOLT-4 (**q**) cells. Cells were treated with indicated Trametinib (0.5 or 1 μM) for 24 h (**p, q**). All IB data are representative of two independent experiments. Source data are provided as a Source data file.

development and fertility. There were no distinguishable general appearance and body weights among *Usp5* cKO and WT mice aged 6-10 weeks (Supplementary Fig. 7c, d). The rate of different *Usp5* genotype offspring was as large as expected according to Mendel's laws of heredity and the observed sex ratio was also as expected (Supplementary Fig. 7e, f). We analyzed the T cell populations from the thymus or spleens of WT or *Usp5* cKO mice. The percentages of CD4⁻CD8⁻ double-negative (DN), CD4⁺CD8⁺ double-positive (DP), CD8⁺ single-positive (CD8SP), and CD4⁺ single-positive (CD4SP) cells of total thymocytes are not significantly different between WT and *Usp5* cKO mice (Supplementary Fig. 7g, h). Moreover, the percentages of naive (CD44$^{low}$CD62L$^{high}$), effector/effector memory (CD44$^{high}$CD62L$^{low}$, effector/EM), and central memory (CD44$^{high}$CD62L$^{high}$, CM) cells of total peripheral T cells isolated from spleens are also not significantly altered between WT and *Usp5* cKO mice (Supplementary Fig. 7i, j). Furthermore, there was no significant change of the Foxp3⁺/CD4⁺ Treg cells from the thymus or lymph node (LN) between WT and *Usp5* cKO mice (Supplementary Fig. 7k, l). These results suggest that Usp5 deficiency does not affect T cell development, the population of memory T cells at basal levels.

However, the expression levels of T cell activation markers, CD69 and CD25, were significantly increased in *Usp5* cKO CD8⁺ T cells, but not in the *Usp5* cKO CD4⁺ T cells, upon stimulation with anti-CD3/CD28 antibodies (Supplementary Fig. 7m–p). The production of effector cytokines (IFN-γ and TNF) and cytotoxic molecule (GzmB) was also significantly elevated in *Usp5* cKO CD8⁺ T cells after treatment with anti-CD3/CD28 antibodies (Fig. 5c–e). In contrast, the production of IFN-γ, TNF, and GzmB is not significantly different between WT and *Usp5* cKO CD4⁺ T cells after stimulation with anti-CD3/CD28 antibodies (Supplementary Fig. 7q–s). To explore whether Usp5 affects the suppressive functions of Treg cells, we co-cultured the effector CD8⁺ T cells (Teff) with WT or *Usp5* cKO Treg cells in vitro and detected the effect of Tregs on the proliferation of Teff using the proliferation marker Ki67. Compared to the group without Treg co-culture, the proliferation of Teff cells was significantly decreased when co-culturing with both WT and *Usp5* cKO Treg cells, suggesting the experimental condition works well (Supplementary Fig. 7t). However, there was no significant difference in the Ki67/CD8⁺ T cells co-culturing with WT compared to those co-culturing with *Usp5* cKO Treg cells (Supplementary Fig. 7t). Moreover, the inhibitory cytokine production of IL10 and TGF-β in WT and *Usp5* cKO Treg cells was also not significantly changed (Supplementary Fig. 7u, v). These results together suggest that *Usp5* ablation in T cells promotes the activation of CD8⁺ T cells, but may not affect the functions of CD4⁺ T cells and Treg cells.

Next, we utilized several subcutaneous mouse tumor models to evaluate how *Usp5* deficiency in T cells affects tumor growth in mice. We subcutaneously transplanted the same number of MC38 colorectal cancer cells into WT and *Usp5* cKO C57BL/6J mice. *Usp5* cKO mice exhibited slower growth of MC38 tumors and prolonger survival than WT mice (Fig. 5f–i). The fluorescence-activated cell sorting (FACS) gating strategy was shown in the Supplementary Fig. 8a. The ratio of tumor-infiltrating CD44⁺/CD8⁺ T cells was around 90%, indicating tumor antigen experience (Fig. 5j). *Usp5* deficiency resulted in reduced PD-1 protein level in tumor-infiltrating CD8⁺ T cells (Fig. 5k, l). The infiltrating CD8⁺ T cells and the ratio of CD8⁺/Treg in the tumor microenvironment were significantly increased in MC38 tumors from *Usp5* cKO mice (Fig. 5m, n). However, there was no significant difference of the infiltrating CD4⁺ T cells and Treg cells in MC38 tumors from WT and *Usp5* cKO mice (Fig. 5o, p, and Supplementary Fig. 8b–d). Similar results were obtained using another syngeneic mouse lewis lung carcinoma (LLC) tumor model. The growth of subcutaneous LLC tumors was significantly suppressed in *Usp5* cKO mice (Fig. 5q–s). The PD-1 expression level, but not other receptors we examined, in CD8⁺ T cells was significantly decreased in LLC tumors derived from *Usp5* cKO mice (Fig. 5t–v, and Supplementary Fig. 8e–g). Moreover, the tumor-infiltrating CD8⁺ T cells, but not CD4⁺ T cells nor Treg cells, was significantly increased in LLC tumors derived from *Usp5* cKO mice (Fig. 5w–y). Together, these results show that *Usp5* deficiency in T cells enhances anti-tumor immunity largely through reducing the PD-1 expression in CD8⁺ T cells.

Furthermore, we tested whether simultaneously inhibiting USP5-mediated PD-1 expression in cancer cells and CD8⁺ T cells could further enhance anti-tumor immunity. To this end, we used the shRNA to knockdown *Usp5* expression in mouse E.G7-OVA lymphoma cells and confirmed that knockdown of *Usp5* dramatically reduced PD-1 protein expression in E.G7-OVA cells (Supplementary Fig. 8h). Subsequently, we subcutaneously grafted the shGFP- and shUsp5-treated E.G7-OVA cells in WT and *Usp5* cKO C57BL/6J mice. The results demonstrated that the growth of shUsp5-treated E.G7-OVA tumors was slower than those of shGFP-treated E.G7-OVA tumors in both WT and *Usp5* cKO C57BL/6J mice (Supplementary Fig. 8i, j). However, the growth of shUsp5-treated E.G7-OVA tumors in *Usp5* cKO mice was the most effectively retarded compared with those tumors derived from the other groups (Supplementary Fig. 8i, j). These results provide evidence that simultaneously suppressing the USP5-mediated PD-1 expression in cancer cells and CD8⁺ T cells might further enhance anti-tumor immunity.

## USP5 inhibition combination with Trametinib or CTLA-4 blockade has an additive effect on suppressing tumor growth

Since the ERK-mediated phosphorylation of PD-1 promotes the deubiquitination and stabilization of PD-1 by USP5, we speculated that combined inhibition of USP5 and ERK activation might have an additive effect on decreasing PD-1 protein abundance compared to single-agent treatment. Indeed, PD-1 protein abundance was dramatically decreased upon the combined treatment with USP5 and MEK inhibitors in Jurkat cells (Supplementary Fig. 9a). Furthermore, the combined treatment of EOAI3402143 with trametinib significantly suppressed tumor growth and improved overall survival compared to single-agent treatment in immunocompetent female BALB/c mice bearing CT26 tumors (Fig. 6a–c, and Supplementary Fig. 9b). Moreover, the combination of EOAI3402143 and trametinib significantly reduced PD-1 expression and elevated the infiltrating cytotoxic CD8⁺

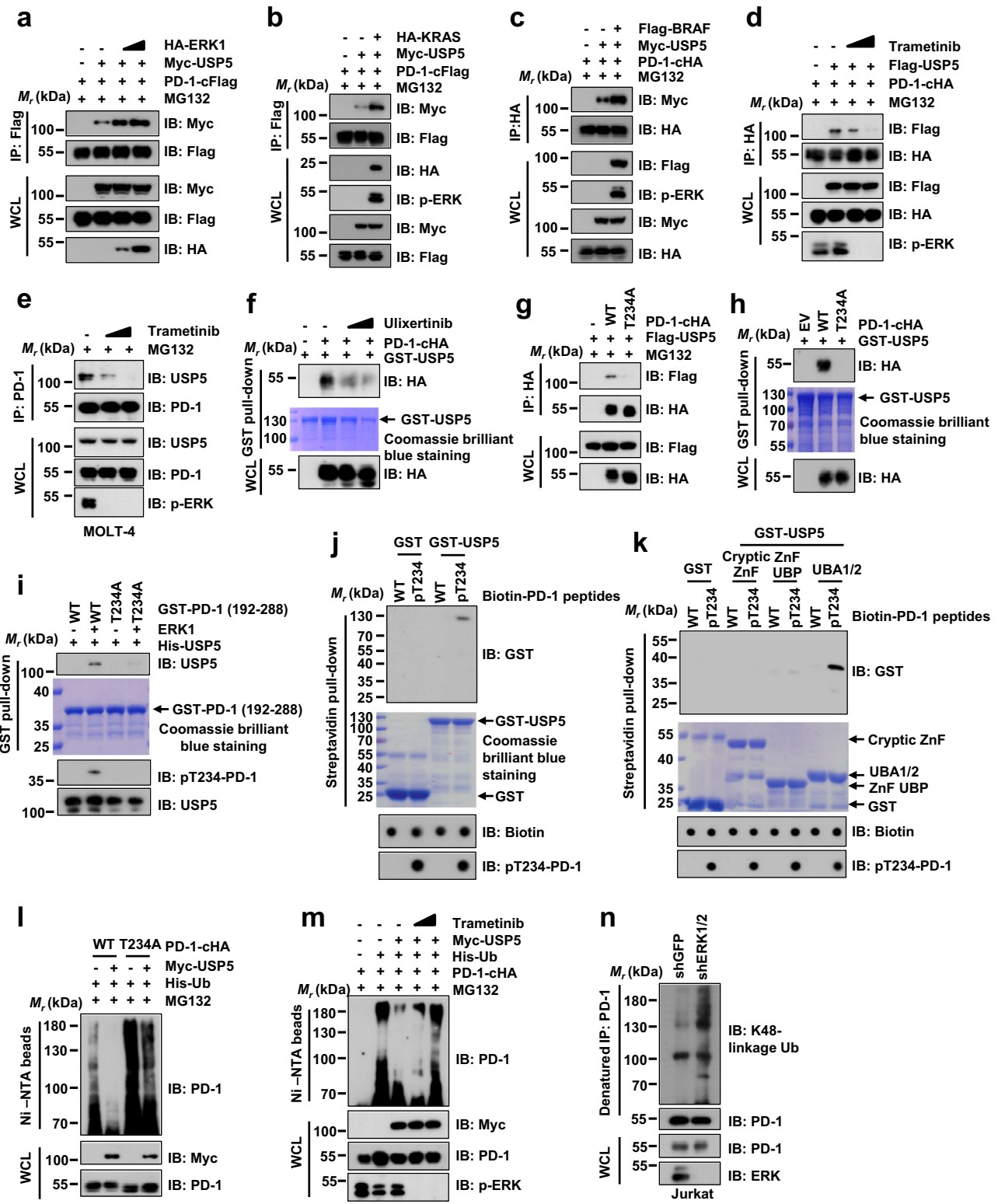

T cells in CT26 tumors (Fig. 6d–f). To evaluate whether the therapeutic efficacy of combined inhibition of USP5 and ERK signaling over anti-PD-1 treatment, we treated the immune-competent female BALB/c mice bearing subcutaneous CT26 tumors. The results showed that the combination of EOAI3402143 and trametinib treatment had better tumor growth control than anti-PD-1 treatment (Supplementary Fig. 9c–f), which might provide a rationale for using EOAI3402143 and trametinib combination treatment over anti-PD-1 therapy in cancer patients.

Furthermore, KP mice bearing autochthonous lung tumors induced by intranasal instillation of Ad-Cre were treated with EOAI3402143, trametinib, alone or combined (Supplementary Fig. 9g). Consistent with the results of syngeneic mouse CT26 tumor model, combined treatment of EOAI3402143 and trametinib significantly decreased PD-1 protein abundance and suppressed tumor development in KP mice, evidenced by reduced tumor sizes and areas (Supplementary Fig. 9h–k). Meanwhile, we did not observe a significant difference in aspects of blood general examination, aspartate amino

**Fig. 4 | ERK-mediated Thr234 phosphorylation of PD-1 promotes its interaction with USP5. a–c** IB analysis of WCL and IPs from HEK293T cells transfected with indicated constructs. Cells were treated with 10 μM MG132 for 12 h. **d, e** IB analysis of WCL and anti-HA IPs from HEK293T cells transfected with indicated constructs (**d**) or anti-PD-1 IPs derived from MOTL-4 cells (**e**). Cells were treated with indicated Trametinib (0.5 or 1 μM) for 24 h and 10 μM MG132 for 12 h (**d**) or 4 h (**e**). **f** IB analysis of WCL and GST pull-down precipitates from HEK293T cell lysates with ectopic expression of PD-1-cHA incubated with recombinant GST-USP5 protein. Cells were treated with 1 μM Ulixertinib for 24 h. **g** IB analysis of WCL and anti-HA IPs from HEK293T cells transfected with indicted USP5, PD-1 WT, or T234A mutant. Cells were treated with 10 μM MG132 for 12 h. **h** IB analysis of WCL and GST pull-down precipitates from HEK293T cell lysates with ectopic expression of PD-1-cHA WT or PD-1-cHA T234A incubated with bacterially purified recombinant GST-USP5 protein. **i** Bacterially purified recombinant GST-PD-1 (192-288) WT or T234A mutant

was phosphorylated by ERK1 in vitro. Subsequently, GST pull-down assay was performed with purified recombinant His-USP5. IB analysis with indicated antibodies. **j, k** 3 μg of indicated biotin-labeled synthetic PD-1 peptides were incubated with 4 μg purified recombinant GST-USP5 (**j**) or different domains (**k**), respectively. Streptavidin beads were added to perform pull-down assays and precipitations were analyzed with IB as indicated. Dot blot was used to identify biotin-labeled synthetic PD-1 peptides. **l, m** IB analysis of WCL and Ni-NTA pull-down products from lysates of HEK293T cells transfected with the indicated constructs and treated with/without Trametinib (1 or 3 μM) for 12 h (**l**). Cells were treated with 10 μM MG132 for 12 h. **n** IB analysis of WCL and anti-PD-1 denatured-IPs from lysates of shGFP- or shERK1/2-treated Jurkat cells using indicated antibodies. Cells were pre-treated with PHA (150 ng/mL) for 3 days and 5 μM MG132 for 6 h. All data are representative of two independent experiments. Source data are provided as a Source data file.

transferase (AST), and alanine amino transferase (ALT) among different treatment groups (Supplementary Fig. 9l–q), suggesting that dosages of inhibitors we used were tolerated in mice. These results together demonstrate that combined inhibition of USP5 and ERK signaling significantly reduces PD-1 protein abundance and activates infiltrating CD8+ T cells to suppress tumor growth in vivo.

Of note, we utilized *Usp5* cKO mice to further test whether Usp5 ablation in T cells also sensitizes tumors to inhibitors of ERK signaling. In keeping with our findings above, *Usp5* cKO mice showed better tumor control than WT mice when treated with Trametinib (Fig. 6g). Moreover, tumors from *Usp5* cKO mice treated with Trametinib displayed reduced PD-1 expression and elevated IFN-γ, TNFα, and GzmB in tumor-infiltrating CD8+ T cells (Fig. 6h–k and Supplementary Fig. 10a, b). However, there was no significant difference of the tumor-infiltrating CD4+ T cells and Treg cells in tumors among different groups (Supplementary Fig. 10c–e). Collectively, these results demonstrate that co-targeting USP5 and ERK signaling has an additive effect on reducing PD-1 and suppressing tumor growth in different preclinical tumor models.

We further evaluated how *Usp5* deficiency in T cells affects the immune checkpoint blockade therapy. To this end, we subcutaneously transplanted the MC38 colorectal cancer cells into *Usp5* WT and cKO C57BL/6J mice, which were treated with anti-PD-1 or control IgG. The anti-PD-1 antibody treatment significantly suppressed tumor growth in WT mice compared with the control IgG treatment (Supplementary Fig. 10f, g). However, there was no significant difference in tumor growth in *Usp5* cKO mice between the control IgG and anti-PD-1 therapy group (Supplementary Fig. 10f, g), indicating that *Usp5* deficiency reducing the PD-1 protein abundance might mimic PD-1 blockade. To further validate this finding, we treated immunocompetent female BALB/c mice bearing syngeneic CT26 tumors with the USP5 inhibitor EOAI3402143, anti-PD-1, alone or combined therapy. The single-agent treatment using EOAI3402143 or anti-PD-1 antibody significantly suppressed the tumor growth and enhanced the overall survival compared with the group of the control treatment (Supplementary Fig. 10h–j). However, the combination of EOAI3402143 and anti-PD-1 treatment did not have an additive role in suppressing tumor growth or elevating the overall survival compared with the single-agent treatment (Supplementary Fig. 10h–j). Taken together, these results suggest that Usp5 inhibition mediated reduction of PD-1 might mimic the PD-1 blockade.

Previous studies have demonstrated that combinational therapy of PD-1 and CTLA-4 blockade is more effective than monotherapy in preclinical mouse tumor models and clinical trials[42–44]. We hypothesized that USP5 inhibition reducing PD-1 expression might improve the efficacy of anti-CTLA-4 immunotherapy in vivo. To test this hypothesis, immunocompetent female BALB/c mice bearing CT26 tumors were treated with the EOAI3402143, the anti-CTLA-4 antibody, alone or combination (Supplementary Fig. 10k). We observed that the treatment with EOAI3402143 plus an anti-CTLA-4 antibody resulted in seven complete responses out of nine treated mice, whereas the anti-

CTLA-4 antibody alone only resulted in two complete responses of the nine treated mice (Fig. 6l). Notably, combination of EOAI3402143 with anti-CTLA-4 therapy significantly improved the overall survival compared with monotherapy (Fig. 6m). In keeping with these findings, the EOAI3402143 treatment alone or combination with anti-CTLA-4 significantly reduced the PD-1 protein abundance in tumor-infiltrating CD3+ T cells and enhanced the production of IFN-γ, TNF, and GzmB in tumor-infiltrating CD8+ T cells compared with control treatment (Fig. 6n–q, and Supplementary Fig. 10l). Together, these results suggest that USP5 inhibition significantly enhances the efficacy of anti-CTLA-4 immunotherapy in vivo, which might be largely due to decreasing PD-1 and increasing cytotoxic activity of tumor-infiltrating CD8+ T cells.

## Discussion

Our study advances the understanding of immune checkpoint PD-1 regulations. We identified USP5 as bona fide deubiquitinase for stabilizing PD-1 through interaction and deubiquitination of PD-1 (Figs. 1 and 2). In addition to ubiquitination, phosphorylation of two tyrosine residues, Y223 in the immunoreceptor tyrosine-based inhibitory motif (ITIM) and Y248 in the immunoreceptor tyrosine-based switch motif (ITSM) in the cytoplasmic region of PD-1, by Lck and/or Src kinase in T cells is critical for PD-1-mediated suppression of T cell receptor (TCR)/CD28 signaling[24,25,45]. This is a tyrosine phosphorylation event distinctive from the Ser/Thr phosphorylation described here. Prior to our studies, whether PD-1 can be Ser/Thr phosphorylated and which kinase(s) phosphorylates Ser/Thr residues on PD-1 is poorly understood.

Our results suggest that the ERK kinase can phosphorylate PD-1 at Thr234, promoting its interaction with USP5 to stabilize PD-1 (Figs. 3 and 4). Based on these mechanisms, we propose a working model that the ERK-mediated phosphorylation on PD-1 at T234 site promotes USP5-mediated deubiquitination of PD-1 and stabilizes PD-1 in CD8+ T cells, resulting in immune evasion and tumorigenesis (Supplementary Fig. 10m, left panel). However, inhibition of USP5 or ERK can reverse this process to destabilize PD-1 and activate CD8+ T cells, resulting in suppressing immune evasion and tumor growth (Supplementary Fig. 10m, right panel).

Lastly, our results show that USP5 inhibition combination with Trametinib or CTLA-4 blockade has an additive role in suppressing the tumor growth (Fig. 6). Although this study has provided enough evidence from biochemical assays and several mouse tumor models to show that USP5 regulates anti-tumor immunity largely through controlling PD-1 protein abundance in CD8+ T cells and affecting the cytotoxic activity of CD8+ T cells, it warrants further in-depth studies to cross the *Pd-1* KO mice with *Usp5* cKO mice to investigate whether the phenotypes of double KO mice are similar with the phenotypes observed in single KO mice. Overall, these data not only define a mechanism for PD-1 regulation by ERK/USP5-mediated phosphorylation and

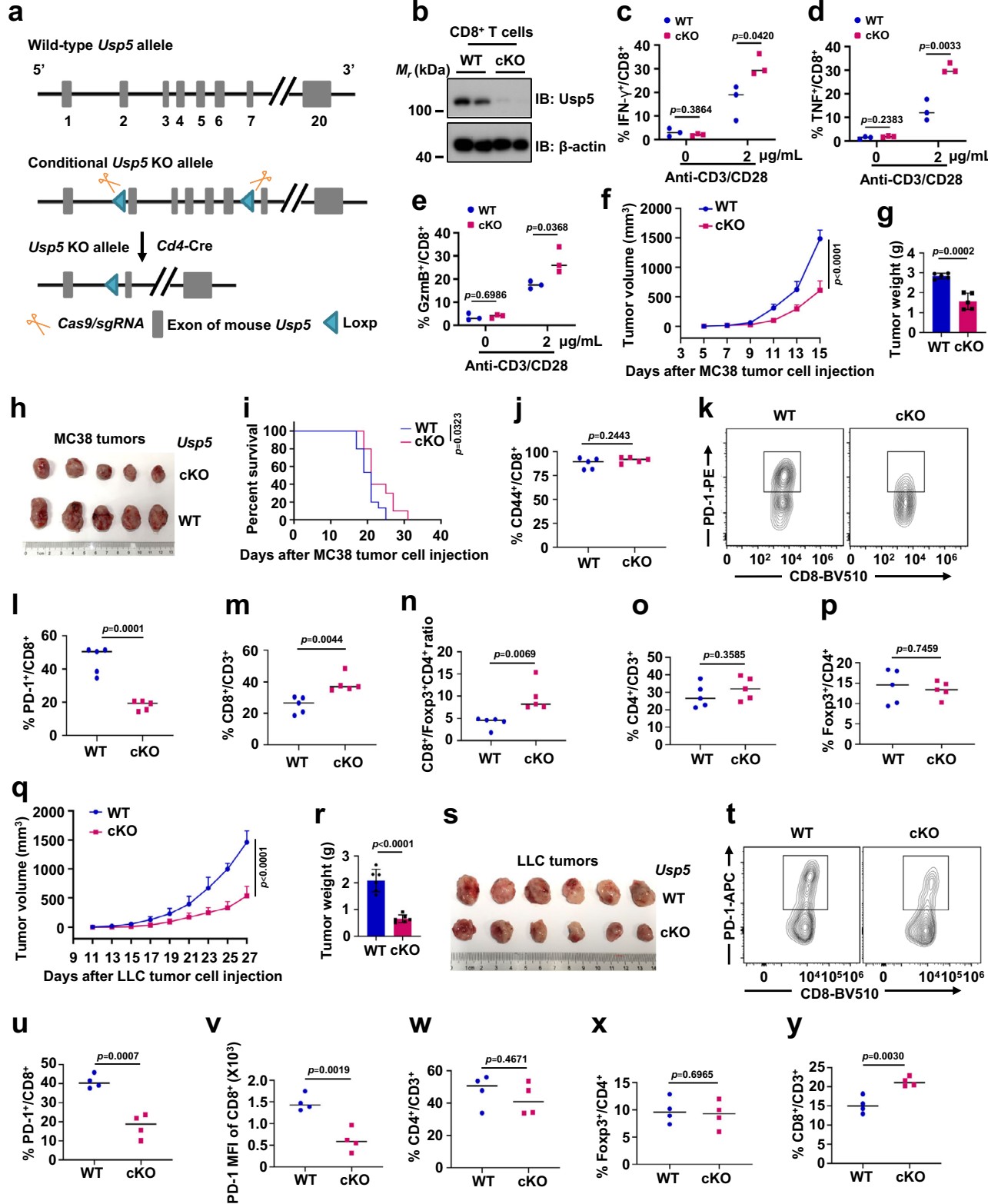

deubiquitination, but also provide potential combined therapeutic strategies for cancer patients.

## Methods
### Mice
Animal studies were approved by the Institutional Animal Care and Use Committee of Wuhan University and complied with all relevant ethical regulations (MRI2021-LAC13, MRI2021-LAC56, and MRI2021-LAC020).

All mice were maintained in pathogen-free facilities at the Medical Research Institute of Wuhan University. 4–6-week-old female BALB/c mice were purchased from Charles River Laboratory. OT-1 TCR transgenic mice were gifts from Dr. Duojiao Wu from Zhongshan Hospital Institute of Clinical Science, Fudan University. *Usp5*flox/flox mice were generated by Cyagen Biosciences on its CRISPR-Cas9 gene targeting platform. *Cd4*-Cre transgenic mice were purchased from Cyagen Biosciences.

**Fig. 5 | *Usp5* conditional knockout (cKO) mice have effective tumor control.**
**a** Conditional knockout strategy of *Usp5*$^{fl/fl}$*Cd4*-Cre (cKO) mice. **b** Protein levels of Usp5 in WT or cKO CD8$^+$ T cells of spleens. **c**–**e** The production of IFN-γ, TNF, or GzmB in WT and cKO CD8$^+$ T cells with anti-CD3/CD28 (0 or 2 µg/mL) stimulation for 96 h and Brefeldin A (BFA, 5 µg/mL) treatment for 5 h. *n* = 3 biologically independent samples. **f**–**i** The growth of subcutaneous MC38 tumors and survival of WT or cKO mice. Tumor growth was measured every 2 days and plotted (**f**). Endpoint tumor weight (**g**) and images (**h**) of MC38 tumors. Kaplan–Meier survival curves for WT and cKO mice bearing MC38 tumors (**i**). Data were presented as mean ± S.D. *n* = 5 mice per group (**f**, **g**). Two-way ANOVA for (**f**). For **i**, WT = 15 mice, cKO = 10 mice, log-rank test. **j**–**l** Quantification of CD44$^+$/CD8$^+$ T cells (**j**) or cellular surface PD-1 on CD8$^+$ T cells (**k**, **l**) in subcutaneous MC38 tumors derived from WT or cKO mice. *n* = 5 mice per group. **m**–**p** Quantification of CD8$^+$/CD3$^+$ T cells (**m**), CD8$^+$/

Foxp3$^+$CD4$^+$ T cells (**n**), CD4$^+$/CD3$^+$ T cells (**o**), or Foxp3$^+$/CD4$^+$ T cells (**p**) in subcutaneous MC38 tumors derived from WT or cKO mice. *n* = 5 mice per group.
**q**–**s** The growth of subcutaneous LLC tumors in WT or cKO mice. Tumor growth was measured every 2 days and plotted (**q**). Endpoint tumor weight (**r**) and images (**s**) of LLC tumors. *n* = 6 mice per group. Data were presented as mean ± S.D. Two-way ANOVA for (**q**). **t**–**v** Flow cytometry analysis of cellular surface PD-1 on CD8$^+$ T cells in subcutaneous LLC tumors from WT or cKO mice (**t**–**v**). Quantification of CD4$^+$/CD3$^+$ T cells (**w**), Foxp3$^+$/CD4$^+$ T cells (**x**), or CD8$^+$/CD3$^+$ T cells (**y**) in subcutaneous LLC tumors derived from WT or cKO mice. *n* = 4 mice per group. For **c**–**e**, **g**, **j**, **l**–**p**, **r**, **u**–**y**, data were presented as mean ± S.D. Two-tailed unpaired *t*-test. For **b**, data are representative of two independent experiments. Source data are provided as a Source data file.

*Usp5*$^{flox/flox}$ mice were generated by conditional knockout (cKO) the exons 2–6 on mouse *Usp5* gene. To generate this mouse, mouse fertilized eggs were injected with a mixture of Cas9, gRNA, and targeting vector. Correctly targeted mice were determined by PCR and gene sequencing. *Usp5*$^{flox/flox}$ mice were then crossed with *Cd4*-Cre transgenic mice to obtain *Usp5* cKO (*Usp5*$^{fl/fl}$*Cd4*-Cre) mice with *Usp5* deficiency in T cells. For animal experiments with 6–8-week-old *Usp5* cKO mice, littermate controls with wild-type (WT) *Usp5* expression (*Usp5*$^{flox/flox}$) were used.

For the following mouse tumor models using WT and *Usp5* cKO mice, both male and female mice were used and were randomly distributed and assigned to each group.

$1.5 \times 10^5$ MC38 or $2 \times 10^6$ LLC cells in 100 µl DMEM were subcutaneously injected into the flank of 6-week-old WT or *Usp5* cKO C57BL/6J mice. On the day of 7 after MC38 tumor cell implantation, mice were treated by intragastric administration of vehicle or Trametinib (0.25 mg/kg) every day for 9 times. Trametinib was dissolved in 5% DMSO, 0.5% Hypromellose (HPMC), and 0.2% Tween 80.

$2 \times 10^5$ shGFP or shUsp5 E.G7-OVA cells in 100 µl DMEM were subcutaneously injected into the flank of 6-week-old WT or *Usp5* cKO C57BL/6J mice.

$1.5 \times 10^5$ MC38 cells in 100 µl DMEM were subcutaneously injected into the flank of 6–8-week-old WT or *Usp5* cKO C57BL/6J mice. On the day of 5 after MC38 tumor cell implantation, mice (5 male/5 female each group) were treated by intraperitoneal (IP) injection of anti-PD-1 (100 µg/mouse) or control IgG every 3 days for 4 times.

For the BALB/c mouse tumor models, only female mice were used. $1.5 \times 10^5$ CT26 cells in 100 µl DMEM were subcutaneously injected into the flank of 6-week-old female BALB/c mice. Mice bearing CT26 tumors were pooled and randomly divided into the indicated groups with comparable average tumor size as indicated treatments in Fig. 6a, Supplementary Figs. 9c, 10k or 10h.

Tumor growth, animal behavior, and mobility were monitored during the treatments. Tumor sizes were measured every 2 days by caliper and tumor volumes were calculated by the formula: length × width$^2$ × 0.5. The maximal tumor burden permitted by the Institutional Animal Care and Use Committee of Wuhan University is 2000 mm$^3$ (MRI2021-LAC13, MRI2021-LAC56, and MRI2021-LAC020). Thus, when tumor volumes reached a maximum of 2000 mm$^3$ or the tumor had ulcers with diameter reached 1 cm, the mice were immediately euthanized.

### Human colorectal cancer (CRC) tissues
CRC tissues from patients were obtained from the Department of Colorectal and Anal Surgery at Zhongnan Hospital of Wuhan University, Wuhan, China. The use of pathological specimens and the review of all pertinent patient records were approved by the Research Ethics Committee of the Zhongnan Hospital of Wuhan University (Protocol # 2020106). Informed consent was obtained by participants. Tissues after surgical resection were immediately stored in 4% paraformaldehyde at 4 °C. These collected surgical

specimens of patients were analyzed by the multiplexed immuno-histochemistry (mIHC).

### Cell culture, transfection, and virus infection
Following cell lines were used in this study: HEK293T (RRID:CVCL_0063), Jurkat (RRID:CVCL_0065), MOLT-4 (RRID:CVCL_0013), CT26 (RRID:CVCL_7254), MC38 (RRID:CVCL_B288), and E.G7-OVA (RRID:CVCL_3505). MC38 was gifted by Dr. Arlene H. Sharpe lab in Harvard Medical School. All other cell lines were originally purchased from American Type Culture Collection (ATCC).

Jurkat, MOLT-4, and E.G7-OVA cells were cultured in RPMI 1640 (Hyclone) medium supplemented with 10% FBS (Gibco). HEK293T cells were cultured in DMEM (Hyclone) medium supplemented with 10% FBS. Mouse CD8$^+$ T cells were cultured in RPMI 1640 medium containing 10% FBS, 10 ng/mL IL-2 (78081, Stemcell), 50 µM 2-mercaptoethanol (M3148, Sigma), and 100 µM non-essential amino acids (11140-050, Gibco). All cells were cultured at 37 °C in an atmosphere of 5% CO$_2$. All media contained 100 units/mL penicillin and 100 µg/mL streptomycin.

HEK293T cells with 70–80% confluence were transfected with indicated constructs using PEI (23966-1, Polysciences) in Opti-MEM medium (Gibco). 36–48 h post-transfection, cells were harvested for various assays including immunoblotting and immunoprecipitation.

HEK293T cells were used to package lentiviruses to generate stable cell lines. For each 10 cm dish, cells were co-transfected with 2 µg pLKO.1-shGFP/target gene, 2 µg VSVG, and 2 µg Δ8.9 plasmids. 48 and 72 h post-transfection, medium with secreted virus particles was harvested and filtered using the 0.45 µm syringe filter. For infection of Jurkat and MOLT-4 cells, cells were first co-incubated with lentiviruses in the presence of polybrene (4 µg/mL, 107689, Sigma) for 90 min through centrifuging with $950 \times g$ at 30 °C. 48 h after infection, cells were selected using puromycin (2 µg/mL for Jurkat or 1.5 µg/mL for MOLT-4) for 3 days. Mouse OT-1 CD8$^+$ T cells were co-cultured with a concentrated lentivirus supplemented with polybrene (10 µg/mL) for 100 min through centrifuging with $950 \times g$ at 30 °C. After 6–8 h in incubator, the lentiviral supernatant was discarded and cells were cultured in complete medium for another 24–36 h. Subsequently, cells were selected using puromycin (5 µg/mL) for 3 days.

### Constructs
pcDNA3.1-PD-1-cHA (Clone ID: OHu26320C) was purchased from Genscript. PD-1-cFlag was subcloned into the pCMV-cFlag vector. Myc-USP5, USP10, USP15, and OTUB1 were generously provided by the laboratory of Dr. Bin Wang from the Daping Hospital in Third Military Medical University. GST-PD-1 truncations, GST-USP5 WT, and truncations were constructed by cloning the corresponding cDNAs into the pGEX-GST-6P vector. HA-tagged kinases used in this study were purchased from Wuhan Miaoling Biotechnology. HA-RAS and Flag-BRAF were generous gifts from Dr. Xiangpeng Dai in the First Hospital, Jilin University. Human PD-1 WT and T234A mutant were subcloned into

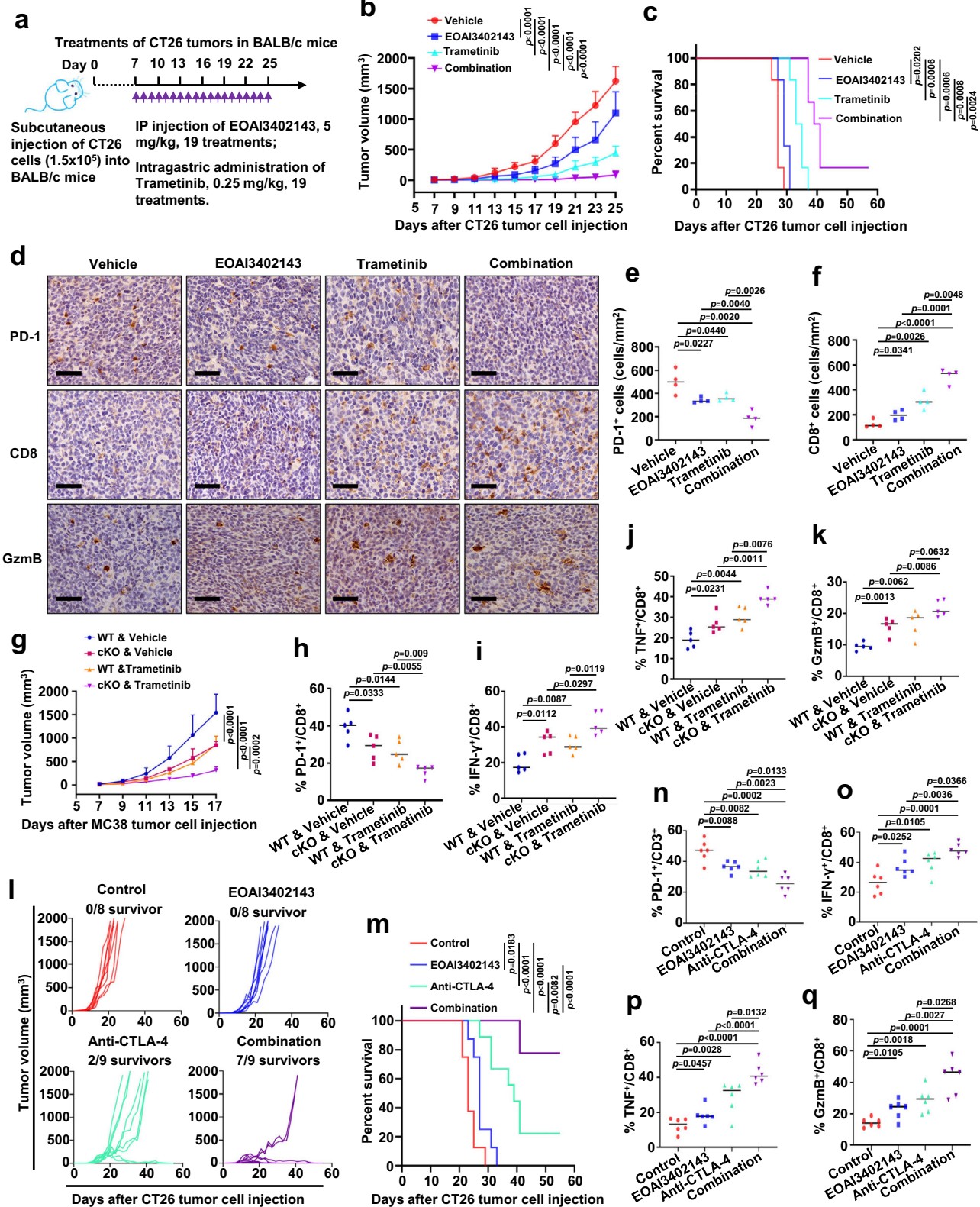

the Retro-pMSCV vector, which was generously provided by Dr.Ying Zhang in the Medical Research Institute, Wuhan University.

The shRNA sequences were inserted into the pLKO.1 vector. Human USP5 shRNAs: #1, 5-GATAGACATGAACCAGCGGAT-3, #2, 5-GACCACACGATTTGCCTCATT-3, #3, 5-CGGGCCACGAACAATAGTTTA-3; mouse Usp5 shRNA: 5-CCCTTAAGTGTTTGCTCCCTT3. human USP10 shRNAs: #1, 5-CGACAAGCTCTTGGAGATAAA-3, #2, 5-CCTAT

GTGGAAACTAAGTATT-3, #3, 5-CCCATGATAGACAGCTTTGTT-3, #4, 5-GCCTCTCTTTAGTGGCTCTTT-3, #5, 5-GCTGTGGATAAACTACCTGAT-3; human USP15 shRNAs: #1, 5-CCCATTGATAACTCTGGACTT-3, #2, 5-CCTTGGAAGTTTACTTAGTTA-3, #3, 5-GCTCTTGAGAATGTGCCGATA-3, #4, 5-GCTCACCAAGTGAAATGGAAA-3; human USP24 shRNAs: #1, 5-TACACTTACCGGGAGTATTTA-3, #2, 5-CTCTCGTATGTAACGTATTTG-3; human USP9x shRNAs: #1, 5-TAAGACAGTGAAT

**Fig. 6 | USP5 inhibition combination with Trametinib or CTLA-4 blockade has an additive effect on suppressing tumor growth. a** A schematic treatment plan for immunocompetent BALB/c mice bearing subcutaneous CT26 tumors. Mice were subcutaneously implanted with $1.5 \times 10^5$ CT26 cells and treated with control vehicle, USP5 inhibitor (EOAI3402143), MEK inhibitor (Trametinib), or combined treatment, respectively. **b, c** The growth of CT26 tumors in BALB/c mice with indicated treatments (**b**). Kaplan–Meier survival curves for BALB/c mice bearing CT26 tumors (**c**). Data were presented as mean ± S.D. Two-way ANOVA (**b**), log-rank test (**c**). $n = 6$ mice per group. **d** Representative images from immunohistochemical (IHC) staining of PD-1, CD8, and GzmB in CT26 tumors from BALB/c mice treated with indicated reagents. Scale bar: 50 μm. **e, f** Quantification for PD-1 (**e**) and CD8 (**f**) positive cells were counted from mice tumor sections. Each plot represents the mean of three random fields of each mouse tumor section. $n = 4$ mice per group. **g** The growth of MC38 tumors in WT or cKO mice with indicated treatment groups. $n = 5$ mice per group. Data were presented as mean ± S.D. Two-way ANOVA.

**h–k** Quantification of flow cytometry result of PD-1⁺ (**h**), IFN-γ⁺ (**i**), TNF⁺ (**j**), and GzmB⁺ (**k**) cells as percentage of CD8⁺ T cells in MC38 tumors after indicated treatments. $n = 5$ each group. **l, m** The growth of CT26 tumors in BALB/c mice with indicated treatments. Tumor volumes of mice treated with control antibody/vehicle ($n = 8$), EOAI3402143 ($n = 8$), anti-CTLA-4 monoclonal antibody ($n = 9$), or combined therapy ($n = 9$) were measured every 2 days and plotted individually (**l**). Kaplan–Meier survival curves for BALB/c mice bearing CT26 tumors (**m**). Data were presented as mean ± S.D. Log-rank test (**m**). **n** Quantification of flow cytometry result of PD-1⁺ as percentage of CD3⁺ T cells in CT26 tumors after indicated treatments. $n = 6$ mice each group. **o, p** Quantification of flow cytometry result of IFN-γ⁺ (**o**), TNF⁺ (**p**), and GzmB⁺ (**q**) cells as the percentage of CD8⁺ T cells in LLC tumors after indicated treatments. $n = 6$ mice each group. For **e, f, h–k, n–q**, data were presented as mean ± S.D. Two-tailed unpaired *t*-test. Source data are provided as a Source data file.

AAACTCTC-3, #2, 5-TTTCATTGGAAGAATCAGGCG-3; human ERK shRNAs: shERK1, 5-CGACCTTAAGATTTGTGATTT-3, shERK2, 5-TGG AATTGGATGACTTGCCTA-3. All point mutants of genes were generated using Cloned Pfu DNA Polymerase (600153, Agilent) according to the manufacturer's instructions and confirmed by DNA sequencing.

## Antibodies

The following is the information of antibodies used in immunoprecipitation (IP) and immunoblotting (IB). Anti-PD-1 (D4W2J) rabbit mAb (1:2000 dilution, Cat# 86163, RRID:AB_2728833), anti-PD-1 (Intracellular Domain) (D7D5W) rabbit mAb (1:2000 dilution, Cat# 84651, RRID:AB_2800041), anti-K48-linkage Specific Polyubiquitin rabbit pAb (1:1000 dilution, Cat# 4289, RRID:AB_10557239), anti-Phospho-p44/42 MAPK (Erk1/2) (Thr202/Tyr204) rabbit mAb (1:2000 dilution, Cat# 4370, RRID:AB_2315112), anti-Myc-Tag (71D10) Rabbit mAb (1:1000 dilution, Cat# 2278, RRID:AB_490778), and anti-USP9X (D4Y7W) rabbit mAb (1:2000 dilution, Cat# 14898, RRID:AB_2798640) were purchased from Cell Signaling Technology. Anti-USP5 rabbit pAb (1:5000 dilution, Cat# A4202, RRID:AB_2765553), anti-USP10 rabbit mAb (1:5000 dilution, Cat# A4454, RRID:AB_2863277), anti-USP24 rabbit pAb (1:5000 dilution, Cat# A20003, RRID:AB_2862911), anti-ERK1/2 rabbit mAb (1:2000 dilution, Cat# A4782, clone: ARC0212, RRID:AB_2863347), anti-rabbit Control IgG (Cat# AC005, RRID:AB_2771930), anti-mouse Control IgG (Cat# AC011, RRID:AB_2770414), and anti-Biotin rabbit mAb (1:3000 dilution, Cat# A20684) was purchased from ABclonal Technology. Anti-Vinculin (VIN-11-5) mouse mAb (1:100,000 dilution, Cat# V4505, RRID:AB_477617), anti-Flag rabbit pAb (1:50,000 dilution, Cat# F7425, RRID:AB_439687), anti-Flag M2 mouse mAb (1:50,000 dilution, Cat# F3165, RRID:AB_259529), anti-HA rabbit pAb (1:50,000 dilution, Cat# H6908, RRID:AB_260070), anti-HA Agarose (Cat# A2095, RRID:AB_257974), and anti-Flag M2 affinity gel (Cat# A2220, RRID:AB_10063035) were purchased from Sigma-Aldrich. Anti-PD-1/CD279 mouse mAb (1:2000 dilution, Cat# 66220-1-Ig, RRID:AB_2881611), anti-PD-1/CD279 rabbit pAb (1:2000 dilution, Cat# 18106-1-AP, RRID:AB_10732952), anti-USP5 mouse mAb (1:5000 dilution, Cat# 66213-1-Ig, RRID:AB_2881604), anti-USP15 rabbit pAb (1:5000 dilution, Cat# 14354-1-AP, RRID:AB_2257148), and anti-GAPDH mouse mAb (1:50,000 dilution, Cat# 60004-1-Ig, RRID:AB_2107436) were purchased from Proteintech. Purified anti-HA.11 Epitope Tag Antibody (1:2000 dilution, Cat# 902301, RRID:AB_2565018) were purchased from Biolegend. Anti-GST tag mouse mAb (Cat# AT0027) was purchased from Engibody.

The rabbit polyclonal phosphorylation antibodies against pT234-PD-1 (1:5000) generated by Abclonal Technology were derived from rabbit with three clones. The antigen sequence for PD-1 used for immunization: 231-REK(p)TPEPP-238. (p)T stands for phosphorylated T234 residue in this synthetic peptide. The antibodies were purified using the antigen peptide column.

The following is the information of antibodies used in flow cytometry analysis. Anti-CD45-APC-Cy7 (1:100 dilution, Cat# 557659, RRID:AB_396774), anti-CD3-BV421 (1:50 dilution, Cat# 562600, RRID:AB_11153670), anti-CD8α-BV510 (1:50 dilution, Cat# 566096, RRID:AB_2739500), anti-CD4-BV605 (1:50 dilution, Cat# 563151, RRID:AB_2687549), anti-PD-1-PE (1:50 dilution, Cat# 551892, RRID:AB_394284), anti-IFN-γ-APC (1:20 dilution, Cat# 562303, RRID:AB_11153140), and anti-TNF-PE (1:20 dilution, Cat# 554419, RRID:AB_395380), anti-CD44-PerCp Cy5.5 (1:50 dilution, Cat# 560570, RRID:AB_1727486), anti-CD25-FITC (1:50 dilution, Cat# 564424, RRID:AB_2738803), anti-CD69-FITC (1:50 dilution, Cat# 553236, RRID:AB_394725), anti-IL10-PE (1:50 dilution, Cat# 554467, RRID:AB_395412), anti-TIM3-BV786 (1:50 dilution, Cat# 747621, RRID:AB_2744187), anti-CD62L-BV650 (1:50 dilution, Cat# 564108, RRID:AB_2738597), anti-Ki67-Alexa Fluor 647 (1:50 dilution, Cat# 561126, RRID:AB_10611874), anti-Foxp3-Alexa Fluor 647 (1:30 dilution, Cat# 560401, RRID:AB_1645201), and fixable viability stain 700 (1:300 dilution, Cat# 564997, RRID:AB_2869637) were purchased from BD Biosciences. Anti-PD-1-AF488 (1:50 dilution, Cat# 329936, RRID:AB_2563594), anti-PD-1-APC (1:50 dilution, Cat# 135210, RRID:AB_2159183), anti-CD28-PerCp Cy5.5 (1:50 dilution, Cat# 102114, RRID:AB_2073850), anti-CTLA-4-PE (1:50 dilution, Cat# 106305, RRID:AB_313254), and anti-TGF-β- PerCp Cy5.5 (1:50 dilution, Cat# 141409, RRID:AB_2561591) were purchased from Biolegend. Anti-GzmB (1:50 dilution, Cat# 61-8898-82, RRID:AB_2574670) was purchased from eBioscience.

The following antibodies were used for multiplexed immunohistochemistry (mIHC), IHC, or immunofluorescence. Anti-human CD3 (1:200 dilution, Cat# 85061, RRID:AB_2721019), anti-mouse CD3 (1:200 dilution, Cat# 99940, RRID:AB_2755035), anti-CK (1:400 dilution, Cat# 4545, RRID:AB_490860), anti-human PD-1 (1:400 dilution, Cat# 86163, RRID:AB_2728833), anti-mouse PD-1 (1:400 dilution, Cat# 84651, RRID:AB_2800041), anti-phospho-ERK1/2 (1:200 dilution, Cat# 4370, RRID:AB_2315112), anti-CD8α (1:400 dilution, Cat# 98941, RRID:AB_2756376) were purchased from Cell Signaling Technology. Anti-Granzyme B (1:800 dilution, Cat# ab4059, RRID:AB_304251), anti-CD4 (1:1500 dilution, Cat# ab183685, RRID:AB_2686917), and anti-Foxp3 (1:1500 dilution, Cat# ab215206, RRID:AB_2860568) were purchased from Abcam. Anti-USP5 rabbit pAb (1:1500 dilution, Cat# A4202, RRID:AB_2765553) and anti-ERK1/2 rabbit mAb (1:200 dilution, Cat# A4782, clone: ARC0212, RRID:AB_2863347) were purchased from ABclonal Technology. Anti-PD-1 (1:400 dilution, Cat# 66220-1-Ig, RRID:AB_2881611), anti-Calnexin (1:50 dilution, Cat# 10427-2-AP, RRID:AB_2069033), anti-Calnexin (1:50 dilution, Cat# 66903-1-Ig, RRID:AB_2882231), and anti-GM130 (1:50 dilution, Cat# 11308-1-AP, RRID:AB_2115327) were purchased from Proteintech.

For tumor mouse model analysis, antibodies we used in this study are below. Anti-mouse PD-1 (100 μg/mouse, clone: 29F.1A12) and anti-

mouse CTLA-4 (100 μg/mouse, clone: 9H10) for treatment were provided by the Laboratory of Dr. Gordon J. Freeman.

## Chemical reagents

EOAI3402143 (HY-111408), Trametinib (HY-10999), Ulixertinib (HY-15816), HPMC (Hypromellose, HY-A0104), PEG300 (HY-Y0873), Tween80 (HY-Y1891), Tunicamycin (HY-A0098), and MG132 (HY-13259) were purchased from MedChemExpress. Cycloheximide (C7698) was purchased from Sigma. Protease inhibitors (B14002) and phosphatase inhibitors (B15001) were purchased from Bimake.

## Immunoprecipitation coupled with mass spectrometry (MS) analysis for identifying PD-1 interacting protein

HEK293T cells transfected with pcDNA3.1-PD-1-cHA or pcDNA3.1-empty vector (EV) were lysed with EBC buffer (50 mM Tris pH 7.5, 0.5% NP40, 120 mM NaCl) supplemented with protease inhibitors (B14002, Bimake) and phosphatase inhibitors (B15001, Bimake). The supernatants of cell lysates were incubated with anti-HA-agarose (A2095, Sigma) in a rotating incubator for 4 h at 4 °C. The resin was washed five times with NETN buffer (20 mM Tris, pH 8.0, 100 mM NaCl, 0.5% NP-40 and 1 mM EDTA). The samples prepared from immunoprecipitation were sent to the MS facility in the College of Life Sciences, Wuhan University. The mass spectrometry analysis and data processing were done by the MS specialist from the MS facility in the College of Life Sciences, Wuhan University.

## Immunoprecipitation (IP) and immunoblot (IB)

The process of IP and IB was described in our previous study[46]. Briefly, cells were lysed in EBC buffer supplemented with protease and phosphatase inhibitors. After quantification of protein concentrations using the BCA protein quantification method, 1–2 mg lysates were incubated with anti-Flag, or anti-HA antibody-conjugated beads, or the anti-PD-1 antibody in a rotating machine for 4 h at 4 °C. After washing four times using NETN buffer, immuno-complexes, and whole-cell lysate were added to 30 μl 3 x loading buffer for boiling for 5 min. The samples were resolved by SDS-PAGE and analyzed by immunoblotting with indicated antibodies.

## Quantitative reverse transcription PCR (qRT-PCR) analysis

Total RNAs from cells were extracted using the TRIzol reagent (15596018, Invitrogen), and reverse transcription reactions were performed using the PrimeScript RT reagent kit (RR047A, TaKaRa). After mixing the cDNA templates with primers and PerfectStart Green qPCR SuperMix (AQ601, TransGen Biotech), reactions were measured by the Bio-Rad CFX Connect Real-Time PCR Detection System (Bio-Rad). The following primers were used for qRT-PCR. For human PD-1: 5-CCAGGATGGTTCTTAGACTCCC-3 (forward), 5-TTTAGCACGAAGCTCTCCGAT-3 (reverse); for human GAPDH: 5-GGAGCGAGATCCCTCCAAAAT-3 (forward), 5-GGCTGTTGTCATACTTCTCATGG-3 (reverse).

## Protein half-life analysis

Cells were treated with 200 μg/mL cycloheximide (CHX) for indicated time points and were harvested for immunoblot analysis using indicated antibodies. The protein band intensity was quantified using the Image J. The quantitative value of the objective protein band was normalized to Vinculin/GAPDH and then compared to the $t = 0$ time point.

## Flow cytometry analysis for cell surface PD-1

The single-cell suspensions were generated in PBS buffer with 1% FBS and 0.1 mM EDTA. Cells were incubated with APC-conjugated anti-human PD-1 antibody (Clone EH12.2H7, Biolegend) for 30 min at 4 °C in the dark, then the cells were washed twice with PBS buffer containing 1% FBS and 0.1 mM EDTA. Flow cytometry data were acquired on a FACS flow cytometer (Beckman, Cytoflex) and analyzed with Flowjo 10.6.2 software (TreeStar).

## GST pull-down assay

Human USP5 WT and different truncates were subcloned into the pGEX-6p-2 construct. Recombinant glutathione S-transferase (GST) tagged USP5 WT and different truncates were purified from *Escherichia coli* BL21. GST pull-down assays were performed by incubating 3 μg GST-USP5 WT or truncates immobilized on the glutathione-Sepharose resin (17-0756-05, GE Healthcare) with HA-PD-1 WT or truncates at 4 °C for 3 h. GST pull-down products were washed five times using NETN buffer and added 2 x loading buffer for boiling 5 min before running SDS-PAGE and analyzing with immunoblotting.

## In vivo deubiquitination assay

The process of in vivo deubiquitination assay is described in our previous study[46]. Briefly, HEK293T cells were co-transfected with indicated constructs including PD-1-HA, His-ubiquitin (His-Ub), Myc-USP5, and other desired constructs. 30 h after transfection, cells were treated with 10 μM MG132 for 12 h. 10% cells were lysed in EBC buffer for input and the remaining cells were sonicated in denaturing buffer A (6 M guanidine-HCl, 0.1 M $Na_2HPO_4/NaH_2PO_4$, and 10 mM imidazole [pH 8.0]) for His-pull down with nickel-nitrilotriacetic acid (Ni-NTA) beads (30230, QIAGEN). After gently rotating for 3 h at room temperature, products of His-pull down were washed twice with buffer A, twice with buffer A/TI (vol:vol = 1:3), and once with buffer TI (25 mM Tris-HCl and 20 mM imidazole [pH 6.8]). Pull-down proteins were resolved with SDS-PAGE for immunoblotting using indicated antibodies.

## In vitro deubiquitination assay

HEK293T cells co-transfected with His-ubiquitin and PD-1-cHA or PD-1-cFlag were treated with 10 μM MG132 for 12 h before harvesting. Ubiquitinated PD-1 was purified by the anti-HA or anti-Flag beads. Ubiquitinated PD-1 were incubated with or without bacterially purified recombinant GST-USP5 WT, GST-USP5 C335A, purified His-USP5 (12772-H08B, Sino biological), or purified Usp5 proteins from mouse primary CD3+ T cells in deubiquitination buffer (60 mM HEPES, 5 mM $MgCl_2$, 4% glycerol, PH 7.6). After incubation for 4–6 h at 37 °C, samples were added to 5 X SDS loading buffer (BL502A, Biosharp) for boiling for 5 min and resolved by SDS-PAGE for immunoblot analysis.

## Denatured IP for endogenous PD-1 ubiquitination assay

For endogenous PD-1 ubiquitination assays, cells were treated with the proteasome inhibitor MG132 as indicated and were lysed in denaturing buffer (50 mM Tris, 1% SDS, 0.5 mM EDTA, and 1 mM dithiothreitol, pH 7.5). After incubation at 100 °C for 10 min, the cell lysates were sonicated and incubated with PD-1 antibodies overnight at 4 °C. 50 μl protein A Sepharose (17078001, Cytiva) was added for another 6 h at 4 °C. The immunoprecipitants were washed with NETN buffer four times and resolved by SDS-PAGE for immunoblotting.

## In vitro kinase assay

Recombinant human GST-PD-1 (23-170), PD-1 (192-288) WT, and T234A mutant proteins were purified from *E. coli* BL21 cells. 0.1 μg human active ERK1 (E7407, Sigma) was incubated with 1 μg recombinant GST or GST-PD-1 truncates in the kinase assay buffer (25 mM Tris-HCl at pH 7.5, 25 mM KCl, 5 mM $MgCl_2$) with 500 μM ATP-γ-S (ab138911, Abcam) and 1 mM DTT for 1 h at 30 °C. The reaction was stopped by adding 2 μl EDTA (0.5 M) and were further incubated with 0.5 mM p-Nitrobenzyl mesylate (PNBM) (ab138910, Abcam) for 1 h at 30 °C. Subsequently, the signal of phosphorylation was detected by immunoblotting with anti-thiophosphate ester antibody (ab92570, Abcam).

## MS analysis for identifying phosphorylation on PD-1

The coomassie-stained SDS-PAGE gel pieces were destained with a destaining solution of 50 mM ammonium bicarbonate in 50% acetonitrile (1:1, vol/vol) and dehydrated in acetonitrile. Gel pieces were rehydrated with 10 mM DTT in 25 mM ammonium bicarbonate and

incubated at 56 °C for 60 min for reduction, followed by alkylation with 55 mM iodoacetamide in 25 mM ammonium bicarbonate for 45 min in the dark. Gel pieces were washed with destaining solution twice and water once, dehydrated in acetonitrile, rehydrated with 0.01 µg/µl trypsin (Promega) in 10 mM ammonium bicarbonate at 0–4 °C for 30 min, and then incubated at 37 °C overnight. Peptides extraction from the gel pieces were performed by sequential incubation with 50% acetonitrile and 100% acetonitrile. Peptides were dried completely by a speed-vac.

The dried peptides were resuspended in 30 µl of LC-MS buffer A (0.1% FA in 2% acetonitrile), and 15 µl was subjected into an UltiMate 3000 UHPLC (Thermo Scientific, San Jose, CA) coupled online with Q Exactive HF-X mass spectrometer (Thermo Scientific). A 150 µm × 30 cm analytical column was packed in-house with 1.7 µm C18 resin. The HPLC gradient was set as follows at a flow rate of 500 nl/min: 5% buffer B (0.1% FA in 98% acetonitrile) over 5 min, 5–25% buffer B over 40 min, 25–35% buffer B over 5 min, 35–80% buffer B over 2 min, 80% buffer B over 2 min, 80–5% buffer B over 0.5 min, and 5% buffer B over 10.5 min.

For data-dependent acquisition (DDA), full-MS ($m/z$ 350–1500) was acquired in the orbitrap with a resolution of 60,000, AGC target of 3e6, and maximum IT of 80 ms. For MS/MS, the top 30 most abundant ions were automatically selected in each MS scan and fragmented in HCD mode at a resolution of 15,000, AGC target of 1e5, and isolation window of 2.0 $m/z$. Normalized collision energy (NCE) was set to 28.

Raw data were processed and searched against the Swiss-Prot human database using MaxQuant (version 1.5.3.0) for protein identification and quantification. Label-free quantification (LFQ) was enabled. Both the PSM FDR and protein FDR were set to 0.01. Other parameters in MaxQuant were set to the default values.

### PD-1 peptides and USP5 binding assays
PD-1 peptides with/without phosphorylation at Thr234 (pT234) were synthesized at GenScript Biotech Corporation. Peptides were diluted to 1 mg/mL with PBS for further biochemical assays. PD-1 WT: Biotin-ELDFQWREKTPEPPVPCV; PD-1 pT234: Biotin-ELDFQWREKT(p) PEPPVPCV. Peptides (3 µg) were incubated with 4 µg bacterially purified recombinant GST-USP5 and three individual domains in a total volume of 500 µl reaction buffer (50 mM Tris pH 7.5, 100 mM NaCl). After incubation for 4 h at 4 °C, 10 µl Streptavidin Agarose (HY-K0218, MedChemExpress) was added to the sample for another 1.5 h incubation. The agarose was washed five times with NETN buffer. Streptavidin Agarose and bound proteins were added to a 2×SDS loading buffer and resolved by SDS-PAGE for immunoblot analysis.

### Multiplex immunohistochemical (mIHC) staining
Tissues with paraformaldehyde-fixed and paraffin-embedded were cut into 2 µm sections. After deparaffinized and rehydrated, the slices were fixed by 4% paraformaldehyde for 20 min and then antigen retrieval as normal IHC. mIHC staining was performed by using the Opal 7-Color Manual IHC Kit (NEL811001KT, Akoya Biosciences). The tumor sections were blocked in Opal Antibody Diluent/Block for 12 min at room temperature. Primary antibodies were incubated for 1 h at 37 °C or overnight at 4 °C. Following washing in TBST, the sections were incubated with secondary Abs Opal Polymer HRP Ms+Rb for 10 min at 37 °C. Sections were then washed in TBST and stained for 10 min with fluorescence staining diluted 1:100 in 1× Plus Amplification Diluent. All the slides were stained with DAPI for 5 min at room temperature and imaged by Akoya Vectra3. The images were sequentially spectrally unmixed by Akoya phenoptics inForm software.

### Immunohistochemical (IHC)
Tissues with paraformaldehyde-fixed and paraffin-embedded were cut into 4 µm sections. Slices were deparaffinized in xylene, rehydrated in 100, 95, and 75% ethanol for 5 min, respectively. Slices were then proceeded to heat-mediated antigen retrieval in 0.01 M citric acid

buffer (pH 6.0) by heating in a microwave oven for 15 min. Subsequently, sections were incubated with 3% hydrogen superoxide for 15 min at room temperature to quench endogenous peroxidase activity. After incubating in 10% normal goat serum for 1 h to block non-specific binding, sections were incubated with primary antibodies overnight at 4 °C. After washing three times in PBS for 5 min each time, sections were incubated with a secondary biotinylated goat immunoglobulin G antibody solution for 1 h. After washing with PBS, specific detection was developed with the Servicebio Detection Kit peroxidase/ diaminobenzidine (DAB) rabbit/mouse (G1212-200T, Servicebio), followed by counterstaining with Mayer's hematoxylin. Immunohistochemical staining was scanned using the Leica Aperio VERSA 8 (Aperio imagescope (v12.3.2.8013) multifunctional scanner. The intensities of DAB staining were measured and quantified using Aperio Quantification software.

### Endoplasmic reticulum (ER) and Golgi apparatus isolation assays
ER and Golgi apparatus microsomal fractions were extracted by ER enrichment kit (ER-036, Invent Biotechnologies, Inc.) and Golgi apparatus enrichment kit (GO-037, Invent Biotechnologies, Inc.). All experiments were performed according to per manufacturer's instructions. The enrichment proteins were resolved by SDS-PAGE and detected with indicated antibodies by IB.

### Immunofluorescence staining for Jurkat or MOLT-4 cells
$3 \times 10^5$ Jurkat or MOLT-4 cells were obtained by centrifugation at $300 \times g \times 5$ min. After washing with phosphate-buffered saline (PBS), the cells were fixed with 50 µl 4% paraformaldehyde (PFA) for 10 min. Fixed cells were washed with PBS by centrifugation at $800 \times g \times 5$ min and were resuspended with 100 µl PBS. Subsequently, cells were seeded onto the slide with Thermo Scientific Shandon Cytospin® 4 Cytocentrifuge at $300 \times g \times 5$ min. The cycle around the cells was drawn using an immunity staining guard pen (BC005, Biosharp). Next, the cells were incubated with 40 µl PBS containing 0.1% Triton X-100 for 10 min. The cells were blocked with 5% bovine serum albumin (BSA), and incubated with primary antibodies dilution at 4 °C overnight. After washing with PBS containing 0.1% Tween 3 times, the slides were incubated with a second antibody Cy3-labeled goat anti-mouse IgG(H + L) (A0521, Beyotime) and Alexa Fluor 488-labeled goat anti-rabbit IgG(H + L) (A0423, Beyotime) avoid of light for 1.5 h. Secondary antibodies were washed with PBS containing 0.1% Tween 3 times. At last, the nucleus was stained with 4′,6-diamidino-2-phenylindole (DAPI) (C0065, Solarbio). The slides were analyzed for the immunofluorescence signal using laser scanning confocal microscopy (LSCM, ZEISS).

### In vitro T-cell-mediated tumor cell-killing assay
CD8+ T cells isolated from spleens of OT-I mice by negative selection magnetic beads (19853, Stemcell Technologies) were stimulated with plate-bound anti-CD3 antibody (100340, Biolegend) and anti-CD28 antibodies (102116, Biolegend) for 48 h. CD8+ T cells were then cultured in RPMI 1640 medium supplemented with 10% FBS, antibiotics (100 units/mL penicillin and 100 µg/mL streptomycin), 10 ng/mL IL-2, 50 µM 2-mercaptoethanol, and 100 µM non-essential amino acids. Subsequently, OT-1 CD8+ T cells were infected with lentiviruses particles expressing shUsp5 for knocking down USP5 as well as shGFP as a negative control. To test the sensitivity of cancer cells to T cell-driven cytotoxicity, we seeded the colon cancer MC38 cells at equal number in twelve-well plates. MC38 cells were incubated with 0.5 µg/mL OVA peptides (S7951, Sigma) for 12 h and then OT-1 CD8+ T cells were added to cancer cells. The ratios of effector (OT-1 CD8+ T cells) to target cells (MC38 cells) were 1:1, 2:1, and 3:1. Apoptotic cells were detected with Annexin V/Propidium Iodide kit (C1067M, Beyotime Biotechnology).

## Mouse primary T cell isolation and effector function analysis

Splenic CD4$^+$ or CD8$^+$ T cells were isolated from the WT or *Usp5* cKO mice by negative selection magnetic beads (B90001 for CD4$^+$ T cell isolation, B90011 for CD8$^+$ T cell isolation, Selleckchem). CD4$^+$ or CD8$^+$ T cells were cultured with 10 ng/mL IL-2 (78081, Stemcell) and stimulated with plate-bound 2 µg/mL anti-CD3 (100340, Biolegend) and anti-CD28 (102116, Biolegend) for 96 h in 48-well plates. To detect cytokine secretion including IFN-γ, TNF, and GzmB, cells were treated with 5 µg/mL Brefeldin A (abs810012, Absin) for 5 h to block cytokine transportation before harvesting. Subsequently, cells were stained with APC-IFN-γ, PE-TNF, and PE-CF594-GzmB for analysis by flow cytometry.

## Tumor-infiltrating lymphocytes isolation for flow cytometry analysis

Freshly resected tumors from mice were first mechanically minced into about 1–3 mm pieces and chemically digested in dissociation buffer (1 mg/mL collagenase IV (BS156, Biosharp), 0.5 mg/mL collagenase I (BS153, Biosharp), 0.002 mg/mL hyaluronidase (BS171, Biosharp) in HBSS) with rotation at 37 °C for 30 min. After passing through a 70 µM strainer, red blood cells in the single-cell suspensions were lysed with ACK lysis buffer (A1049201, Thermo Fisher). To analyze T cell effector function, cells were incubated with appropriate antibodies for 30 min at 4 °C in dark. Flow cytometry was performed on FACSCelesta (BD Biosciences), and data were analyzed using Flowjo 10.6.2 software (TreeStar).

## In vitro Tregs suppression assay

Wild-type (WT) CD8$^+$ T cells were co-cultured with WT or *Usp5* cKO CD4$^+$CD25$^+$ T cells for 4 days in the presence of plate-bound 1 µg/mL anti-CD3/CD28. The suppressive activity of CD4$^+$CD25$^+$ T cells was determined by measuring the proliferation of activated CD8$^+$ T cells staining with APC-Ki67. At the same time, the inhibitory cytokine production of IL10 and TGF-β in WT and *Usp5* cKO CD4$^+$CD25$^+$ T cells was also measured.

## *Kras*$^{LSL-G12D/+}$*Trp53*$^{fl/fl}$ (KP) mice tumor model

*Kras*$^{LSL-G12D/+}$*Trp53*$^{fl/fl}$ (KP) mice were kindly provided by the laboratory of Dr. Bo Zhong at Wuhan University. For treatments with combined USP5 inhibitors and MEK inhibitors in the lung cancer KP mice tumor model, the treatment plan was performed as illustrated in Supplementary Fig. 9g. Briefly, seven-week-old KP mice were anesthetized by intraperitoneal injection of 1% sodium pentobarbital (w/v = 1:7), followed by intranasal injection of Adenovirus-Cre viruses for $2 \times 10^6$ pfu in 50 µl PBS per mouse (Obio Technology). Thirty-five days after infection, mice were pooled and randomly divided into the following four groups: (1) control vehicle treatment, (2) EOAI3402143 treatment, (3) Trametinib treatment, and (4) combined treatment. At the indicated time points after treatment, mice were euthanized and the samples of lung and whole blood were harvested for subsequent analysis. The mice sex was randomly distributed and assigned to each group.

## Hematoxylin-eosin (HE) staining analysis

Lung tissues from *Kras*$^{LSL-G12D/+}$*Tp53*$^{fl/fl}$ (KP) mice were fixed in 4% paraformaldehyde and embedded into paraffin blocks. The paraffin blocks were sectioned (5 µm) for Hematoxylin-eosin staining (Beyotime Biotech) followed by coverslipped. Images were acquired using an Aperio VERSA 8 (Leica) multifunctional scanner.

## Statistical analysis

Statistical analysis was performed in Prism 8.0 (GraphPad) software. Quantitative data were presented as mean ± SD, as indicated by at least three independent experiments by Student's *t* test among two group differences. Growth curves of tumor volumes were analyzed using two-way ANOVA test. For animal survival analysis, the Kaplan–Meier method was adopted to generate graphs, and the survival curves were analyzed with log-rank test. *P* values < 0.05 were considered statistically significant.

## Reporting summary

Further information on research design is available in the Nature Portfolio Reporting Summary linked to this article.

## Data availability

The mass spectrometry proteomics data for identifying the potential PD-1-interacting proteins have been deposited to the ProteomeXchange Consortium (http://proteomecentral.proteomexchange.org) via the iProX partner repository with the dataset identifier PXD040671. The remaining data are available within the Article, Supplementary Information or Source data file. Source data are provided with this paper.

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

## Acknowledgements

This work was supported by grants from the National Key Research and Development Program of China (2022YFC3401500 to J.Z.), the National Natural Science Foundation of China (31970732 and 82273062 to J.Z., 82203319 to M.G., 82103149 to H.Y.) and the Fundamental Research Funds for the Central Universities (2042022dx0003 to J.Z. and H.Y.), Development Project of Hubei Province (2020BCA069 to C.X), Knowledge Innovation Program of Wuhan-Basic Research (J.Z.), the Natural Science Foundation of Hubei Province of China (2022CFA008 to J.Z.), translational Medicine and Interdisciplinary Research Joint Fund of Zhongnan Hospital of Wuhan University (ZNJC201922 to C.X. and J.Z.), Research Project established by Chinese Pharmaceutical Association Hospital Pharmacy department (CPA-Z05-ZC-2022-002 to J.Z.), and the Engineering construction project of improving diagnosis and treatment ability of difficult diseases (oncology) (2022CFA008 to C.J.). This work was also supported by the National Cancer Institute of USA (P50CA101942) to G.J.F. We thank the laboratory of Dr. Shengdian Wang (Institute of Biophysics, Chinese Academy of Sciences) for generously providing the E.G7-OVA cell line. We also thank the staff at the core facility of the Medical Research Institute at Wuhan University for their technical support.

## Author contributions

J.Z., C.X., Xiangling.X., J.S., and C.H. designed research; Xiangling.X., J.S., and C.H. performed most of the experiments with help from Y.S., M.G., B.X., W.X., P.D., Q.M., Xixin X., Y.Y., H.Y., and G.X.; X.B. and G.J.F. provided essential reagents; S.L. and Y.R. performed the mass spectrometry experiments for identifying the ubiquitination and phosphorylation on PD-1. B.C. and C.J. collected human colorectal cancer tissues. Geng M. analyzed the structure of USP5 UBA1 and UBA2 domains. J.Z. and C.X. guided and supervised the project. J.Z., Xiangling.X., and M.G. wrote the manuscript. W.W., G.J.F., X.B., Y.-R.L., and C.X. gave discussion and edited the manuscript. All authors commented on the manuscript.

## Competing interests

G.J.F. has patents/pending royalties on the PD-1/PD-L1 pathway from Roche, Merck MSD, Bristol-Myers-Squibb, Merck KGA, Boehringer-Ingelheim, AstraZeneca, Dako, Leica, Mayo Clinic, Eli Lilly, and Novartis. G.J.F. has served on advisory boards for Roche, Bristol-Myers-Squibb, Xios, Origimed, Triursus, iTeos, NextPoint, IgM, Jubilant, Trillium, GV20, IOME, and Geode. G.J.F. has equity in Nextpoint, Triursus, Xios, iTeos, IgM, GV20, Invaria, and Geode. W.W. is a co-founder and consultant for ReKindle Therapeutics. The remaining authors declare no other competing interests.
