## [Peer Review File · Nature Communications]

ERK and USP5 govern PD-1 homeostasis via deubiquitination to modulate tumor immunotherapyREVIEWER COMMENTS

Reviewer #1 (Remarks to the Author): with expertise in PD1/PDL1 regulation, cancer immunology

In this study, Xiao et al. discovered a post-translational modification of PD-1 and its impact on regulating T cell activity against cancers. The authors identified a deubiquitinase, USP15, controlling PD-1 homeostasis through protein stabilization. Mechanistically, ERK phosphorylates PD-1 at Thr234 and recruits USP5 to stabilize PD-1, resulting in T cell exhaustion. In this regard, simultaneous inhibition of ERK and USP15 activation may reduce PD-1 expression with promising therapeutic efficacy. This study identified a novel deubiquitinase of PD-1 with a comprehensive biochemical analysis. The authors generated site and phospho-specific mAb and further used conditional ko mice to demonstrate the anticancer activity of USP15 in vivo. Overall, this is a well-organized manuscript with a few issues remaining to be addressed before being accepted for publication.

1. N-linked glycosylation of PD-1 at N49, N58, N74, and N116 is known to regulate its protein localization and stability. It is unclear if ERK/USP5-mediated PD-1 stabilization happened before or after protein glycosylation.
2. Does ERK also express in the ER? Since PD-1 is a glycoprotein, the majority of the protein is located in the ER and Golgi. It is not clear where ERK/USP5 signaling happened in the T cells.
3. How effective is the EOAI3402143 and trametinib combination in immune-competent mice compared to anti-PD-1 treatment? The authors need to provide a rationale for using the EOAI3402143 and trametinib combination over anti-PD-1.
4. The gating of Fig. 3F is weird. It looks like the author tried to gate the area that matches the hypothesis. It isn't objective.
5. The authors could draw an illustration to describe the mechanism.
6. They should provide initial mass spectrometry data as well as immunoprecipitation data.

Reviewer #2 (Remarks to the Author): with expertise in T cell biology, cancer immunology

The paper by Xiao et al documents the ability of USP5 to regulate PD1 expression via ubiquitination. It is an interesting paper of potential importance but the description of the figures is often lacking detail and many experiments lack controls. The majority of experiments were conducted in over expression systems in transformed cells lines and so it is unclear the degree to which events occur in primary T-cells. They need to demonstrate the functional importance of the ERK-USP5 pathway in regulating PD1 function in primary T-cells.

Specific points

- 1) The authors should change the title to include a reference to ubiquitination.
- 2) Figure 1: Panel a: more detail is needed on the validity of the screen. Several positive candidates appear to be false positives? Why was this? Panel b: is unclear what the expression levels are for myc-U5PS. Panel d seems a real problem. No information is provided on nature of ?? Panel f requires more information on the nature of the shRNA and its specificity. There is no control for the possible effect of the shRNA on a related USP. Panel J: the effect of EQAI3402143 is impressive; however again this is an over expression system. The authors need to conclude data on primary T-cells and also to include evidence that the enzymatic activity of USP5 is directly inhibited. They need to isolate the protein and run in vitro assays. Panel q is inadequate. Real numbers are needed as well as a definition of what is considered overlapping or which involves quenching.
- 3) Figure 2: Panel A: Jurkat cells are abnormal lacking PTEN and other key enzymes. It is confusing since Jurkat cells generally express little PD1. Are the authors transfecting with PD-1 in these cells? The co-IP is convincing but needs to be convincingly demonstrated in activated primary mouse and human T-cells expressing PD1. Panels c-e: deletion analysis is fine as a first step but large deletions can have indirect effects due to conformation changes or changes in interactions with the membrane

or associated proteins It would strengthen the paper greatly if the authors could identify some specific residues (within motifs) needed for the interaction and to use these single or double mutated proteins to demonstrate effects. Panel h-l: convincing but confusing. PD1 has been worked on for over a decade and it generally migrates as a single band on a gel with little evidence of the smear indicative of ubiquitination in primary cells- is this not the case? It may be that the over expression of protein is somehow promoting the need for its degradation in a manner not normally seen in primary T-cells? A clear demonstration of PD1 Ub in normal T-cells is needed combined with a clear demonstration of the effects of USP5 in the KO primary T-cells.

4) Figure 3: Panel a: PD1 flag is phosphorylated by several kinases in this assay system of over expression. ERK is just one of several kinases including AKT. The authors need to show specificity for ERK by titrating the kinase levels and to use sh/siRNA to KD ERK showing a loss of phosphorylation in primary T-cells. Does the single site phosphorylated when mutated, abrogate the effects of USP5 in modulating PD-1 function in primary T-cells? Is ERK more effective at lower levels of expression relative to the other kinases? Panel d is not adequate, numbers and sample size criteria are needed.

5) The authors also need to numbers of photo cuts of gels to allow the reader to see whether other bands exist or not..

6) Figure 5: the generation of the KO mice is good. They now need to use the primary T-cells from these mice to show PD1 ubiquitination, ERK driven phosphorylation and the preferential effects of the loss of USP5 on PD-1 function and checkpoint blockade in these mice. Further in Panel e, although the effect is consistent with its importance; the authors need to include anti-PD1 treatment as a control- presumably anti-PD-1 may be more or less impactful due to altered PD1 expression? They also need to include several tumor models and some information on the nature of the altered expression of other receptors such as CD28, CTLA-4, TIM3 etc. as well as a nature and definition of memory vs effector memory and effector subsets. The authors should also consider crossing pd1^{-/-} mice with their Usp5 KO mice where the phenotypes should be the same as the phenotypes seen in single KO mice?

5) The description of the figures in the text is wholly inadequate. There is hardly any description of how experiments were done or text to guide the reader. One is just expected to accept a figure of results with a single line of description.

Reviewer #3 (Remarks to the Author): with expertise in ubiquitination, immunology

Xiao et al identified USP5 as a PD-1-specific deubiquitinase using an IP-MS proteomic approach. They further discovered that Erk-mediated PD-1 phosphorylation facilitates USP5-PD-1 interaction and USP5-mediated PD-1 stabilization. Targeted deletion either in lymphoma cells by shRNA or conditional KO in mouse T cells, or pharmacological USP5 inhibition reduced PD-1 expression. As a consequence, inhibition of USP5 in T cells resulted in enhanced antitumor immunity. Overall, this is a novel study, and the data largely document their main conclusion, but with few concerns to be clarified.

1. The authors generated the T cell specific USP5 conditional KO mice, but their phenotype is under characterized. Since PD-1 is expression in all T cell subsets including CD4, CD8 effector cells and Tregs, the impact of USP5 gene deletion to CD4 and CD8 T cell activation and Treg development and suppressive functions should be characterized. In particular, it is important to study whether USP5 impairs Treg suppressive functions to document their conclusion that the elevated antitumor immunity is due to the reduced CD8 PD-1 expression.

2. Line up with point #1, the intratumoral CD4 T cells and Tregs should be characterized in Fig. 5 and 6.

3. It will be interesting to test whether to simultaneously inhibit USP5-mediated PD-1 expression in cancer and CD8 T cells using PD-1+ cancer cells, such as EG7 lymphoma, further enhances

antitumor immunity.

4. The authors used a pan-Dub inhibitor EOAI3402143 that inhibits USP9X and USP24 in addition to USP5 for their studies, while the authors argue that USP9X and USP24 inhibition did not affect PD-1 expression, studies to determine whether this inhibitor alters other checkpoint receptors such as PD-L1 should be performed.

5. Data In figure 3e, and figure 4d show that treatment with Trametinib inhibits USP5 expression and overexpression ERK increases USP5 expression as in figure 4a. The authors need to further clarify whether the elevated USP5-PD-1 interaction is due to increased USP5 expression in addition to PD-1 phosphorylation.

6. In figure 4e as well as in 3q, PD-1 appears to be increased in WCL when treated with Erk inhibitor, which inhibited PD-1-USP5 interaction. The authors need to validate this data or explain the discrepancy.

7. The changes in PD-1 MFI on the surface of CD8 T cells in figure 6e and 6h are better to be provided.

8. Several studies have detected that USP5 largely localize in nuclear, but data in Fig 1q show the exclusive USP5 cytoplasmic distribution, which should be validated in a better resolution and discussed accordingly.

Point-by-point response to reviewers' comments
(NCOMMS-22-35135-T)

We sincerely appreciate the thorough analyses and constructive suggestions provided by the three reviewers, which have been very helpful in guiding us to further improve our study. With extensive revision during the past four and half months, as described in more detail below, we have obtained a substantial amount of new experimental evidence, including **33** new main figure panels and **62** new supplementary figure panels of a total of 104 main figure panels and 129 supplementary figure panels, to further strengthen our study. After reading the following point-by-point response, we hope the editor and the reviewers will concur with us that we have experimentally addressed all the concerns raised by three reviewers, and that the revised manuscript is now suitable for publication as a research article in *Nature Communications*.

Reviewer #1 (Remarks to the Author): with expertise in PD1/PDL1 regulation, cancer immunology

In this study, Xiao et al. discovered a post-translational modification of PD-1 and its impact on regulating T cell activity against cancers. The authors identified a deubiquitinase, USP5, controlling PD-1 homeostasis through protein stabilization. Mechanistically, ERK phosphorylates PD-1 at Thr234 and recruits USP5 to stabilize PD-1, resulting in T cell exhaustion. In this regard, simultaneous inhibition of ERK and USP5 activation may reduce PD-1 expression with promising therapeutic efficacy. This study identified a novel deubiquitinase of PD-1 with a comprehensive biochemical analysis. The authors generated site and phospho-specific mAb and further used conditional ko mice to demonstrate the anticancer activity of USP15 in vivo. Overall, this is a well-organized manuscript with a few issues remaining to be addressed before being accepted for publication.

Response: We sincerely thank the reviewer for recognizing that the originally submitted version of our study is a well-organized manuscript. We also thank the reviewer for acknowledging our efforts in providing comprehensive biochemical and mouse modeling evidence to support our major conclusion. At the same time, we thank the reviewer very much for raising the following concerns and insightful suggestions, which have significantly improved our study. Below please kindly find the detailed point-by-point response to each of the reviewer's concerns.

1. N-linked glycosylation of PD-1 at N49, N58, N74, and N116 is known to regulate its protein localization and stability. It is unclear if ERK/USP5-mediated PD-1 stabilization happened before or after protein glycosylation.

Response: We thank the reviewer for raising this great question. As the reviewer mentioned, Sun L, et al., reported that N-linked glycosylation of PD-1 at N49, N58, N74, and N116 is critical for maintaining PD-1 protein stability and localization (**Sun L, et al. *Cancer Res.* 80(11):2298-2310**). To address this concern, we first generated the PD-1 4NQ mutant with substitution of all four sites

asparagine (N) with glutamine (Q) (N49Q/N58Q/N74Q/N116Q) and found that the PD-1 4NQ mutant reduced the molecular weight of PD-1, suggesting that PD-1 lost the N-linked glycosylation (**Revised Supplementary Fig. 5h**), which is consistent with the previous report (**Sun L, et al. *Cancer Res.* 80(11):2298-2310**). The signal for T234 phosphorylation still could be detected on the PD-1 4NQ mutant with the anti-pT234-PD-1 antibody, indicating that missing the N-linked glycosylation on PD-1 does not affect the phosphorylation of PD-1 at the T234 site (**Revised Supplementary Fig. 5h**). However, although the PD-1 T234A mutant lost its phosphorylation, the glycosylation on PD-1 T234A mutant can be detected comparably as the wild-type (WT) PD-1, suggesting that the phosphorylation deficiency at T234 on PD-1 may not affect its glycosylation (**Revised Supplementary Fig. 5h**).

Additionally, treatment with the inhibitor of N-linked glycosylation, tunicamycin, could reduce the N-linked glycosylation on both PD-1 WT and the T234A mutant (**Revised Supplementary Fig. 5i**). Moreover, the tunicamycin-induced un-glycosylation form of PD-1 WT, but not the T234A mutant, can be phosphorylated at the T234 site (**Revised Supplementary Fig. 5i**).

Taken together, these results demonstrate that the N-linked glycosylation and phosphorylation at T234 on PD-1 might not affect each other and should be two parallel pathways to regulate PD-1 stability and localization at different physiological or pathological conditions. We have included all these newly generated data and discussed them in the revised manuscript.

2. Does ERK also express in the ER? Since PD-1 is a glycoprotein, the majority of the protein is located in the ER and Golgi. It is not clear where ERK/USP5 signaling happened in the T cells.

Response: We thank the reviewer for pointing out this critical concern. To address this concern, we first performed the immunofluorescence (IF) and found that ERK colocalized with the endoplasmic reticulum (ER) marker, Calnexin, in the cytoplasm of MOLT-4 cells, suggesting that ERK might localize in the ER (**Revised Supplementary Fig. 5c**). Moreover, PD-1 is located on in the ER and Golgi apparatus, evidenced by its colocalization with the ER maker, Calnexin, and Golgi marker, GM130, in the cytoplasm of MOLT-4 cells (**Revised Supplementary Fig. 5d-g**). Of note, knockdown of *USP5* or *ERK* by shRNAs dramatically decreased the PD-1 protein expression, which might be due to increased ubiquitination and subsequent degradation of PD-1, leading to disrupting its localization in the ER and Golgi apparatus (**Revised Supplementary Fig. 5d-g**).

In addition, we isolated the ER or Golgi from MOLT-4 cells using the ER or Golgi enrichment kit. We detected by western blot and confirmed that ERK, USP5, and PD-1 existed in both the ER and Golgi apparatus albeit the protein level of USP5 and PD-1 was relatively low in the Golgi apparatus compared to ER (**Revised Supplementary Fig. 5a, b**).

Overall, these results suggest that the ERK/USP5 signaling pathway might regulate PD-1 stability to affect its localization on ER and Golgi apparatus in the T cells. We have included all these newly obtained results in revised manuscript.

3. How effective is the EOAI3402143 and trametinib combination in immune-competent mice compared to anti-PD-1 treatment? The authors need to provide a rationale for using the EOAI3402143 and trametinib combination over anti-PD-1.

Response: We thank the reviewer for providing us with this great suggestion. As the reviewer kindly suggested, we treated the immune-competent BALB/c mice bearing subcutaneous CT26 tumors by using anti-PD-1 or the combination of EOAI3402143 and trametinib (**Revised Supplementary Fig. 9c**). Although both anti-PD-1 and the combination of EOAI3402143 with trametinib could significantly retarded tumor growth, the combination of EOAI3402143 and trametinib treatment had better tumor growth control than anti-PD-1 treatment in immune-competent BALB/c mice (**Revised Supplementary Fig. 9d-f**). This result provides a rationale for using the EOAI3402143 and trametinib combination treatment over anti-PD-1 therapy in immune-competent mice. We have included these results in the revised manuscript.

4. The gating of Fig. 3F is weird. It looks like the author tried to gate the area that matches the hypothesis. It isn't objective.

Response: We thank the reviewer for pointing out this concern. We have repeated this assay and used the unstained Jurkat cells as negative control to gate the area and confirmed that knockdown of *ERK1/2* significantly decreased the surface PD-1 expression on Jurkat cells (**Revised Fig. 3f, g**).

5. The authors could draw an illustration to describe the mechanism.

Response: We thank the reviewer for providing us with this great suggestion. As kindly suggested, we have drawn an illustration to describe the mechanism that USP5/ERK regulates PD-1 stability in a phosphorylation and deubiquitination-dependent manner as well as to highlight the combined therapeutic strategies to inhibit immune evasion and tumorigenesis (**Revised Supplementary Fig. 10m**).

6. They should provide initial mass spectrometry data as well as immunoprecipitation data.

Response: We thank the reviewer for providing us with these great suggestions. As kindly suggested, we have deposited the initial mass spectrometry data to the ProteomeXchange Consortium (<http://proteomecentral.proteomexchange.org>) via the iProX partner repository with the dataset identifier [PXD040671](https://proteomecentral.proteomexchange.org/dataset/PXD040671). We also provided the immunoprecipitation data and all other original results in the file of source data.

Reviewer #2 (Remarks to the Author): with expertise in T cell biology, cancer immunology

The paper by Xiao et al documents the ability of USP5 to regulate PD1 expression via ubiquitination. It is an interesting paper of potential importance but the description of the figures is often lacking detail and many experiments lack controls. The majority of experiments were conducted in over expression systems in transformed cells lines and so it is unclear the degree to which events occur in primary T-cells. They need to demonstrate the functional importance of the ERK-USP5 pathway in regulating PD1 function in primary T-cells.

Response: We sincerely thank the reviewer for acknowledging that our study is interesting and potentially important. We also thank the reviewer very much for pointing out concerns and providing us with insightful suggestions, which helps us further significantly improve our study. We fully agree with the reviewer that we should provide more experimental evidence to show the functional importance of the ERK-USP5 pathway in regulating PD-1 function in primary T-cells. As the reviewer kindly instructed, we have thoroughly revised our manuscript and experimentally addressed the concerns raised by the reviewer as listed in our point-by-point responses below.

Specific points

1) The authors should change the title to include a reference to ubiquitination.

Response: Great suggestion. We have changed the title “ERK and USP5 govern PD-1 homeostasis to modulate tumor immunotherapy” into “ERK and USP5 govern PD-1 homeostasis **via deubiquitination** to modulate tumor immunotherapy” in the revised manuscript.

2) Figure 1: Panel a: more detail is needed on the validity of the screen. Several positive candidates appear to be false positives? Why was this? Panel b: is unclear what the expression levels are for myc-USP5. Panel d seems a real problem. No information is provided on nature of ? Panel d requires more information on the nature of the shRNA and its specificity. There is no control for the possible effect of the shRNA on a related USP. Panel J: the effect of EOAI3402143 is impressive; however again this is an over expression system. The authors need to conclude data on primary T-cells and also to include evidence that the enzymatic activity of USP5 is directly inhibited. They need to isolate the protein and run in vitro assays. Panel q is inadequate. Real numbers are needed as well as a definition of what is considered overlapping or which involves quenching.

Response: We thank the reviewer for pointing out these concerns and providing us with insightful suggestions. As instructed, we have provided more information and performed the following experiments to address the concerns for Figure 1:

Panel a: For the screen, HEK293T cells were transfected with pcDNA3.1-PD-1-cHA as well as the pcDNA3.1-cHA empty vector (EV) as control. After 36 hours post-transfection, cells were harvested for immunoprecipitation (IP) using anti-HA-agarose, respectively. We first confirmed that PD-1 was immunoprecipitated in the cell lysates of transfection with pcDNA3.1-PD-1-cHA

construct, but not in the cell lysates of transfection with pcDNA3.1-cHA EV (**Revised Supplementary Fig. 1a**). Subsequently, the immuno-complex from cell lysates expressing PD-1 or EV was analyzed with mass spectrometry (MS) for identifying potential PD-1-interacting proteins. Two reported PD-1-interacting proteins, the ubiquitin E3 ligase adaptor protein KLHL22 and the tyrosine phosphatase SHP2 encoded by the PTPN11 gene, were identified in the anti-HA immuno-complex from cell lysates expressing PD-1, but not in the immuno-complex from cell lysates expressing EV (**Revised Fig. 1a and Supplementary Table 1**). Moreover, the scaffold protein Cullin 3, interacting with KLHL22 and RBX1 forming a functional E3 ligase complex, was also identified in the anti-HA immuno-complex from cell lysates expressing PD-1, but not in the immuno-complex from cell lysates expressing EV (**Revised Fig. 1a and Supplementary Table 1**). These previously identified PD-1-associated proteins confirmed that our screen for identifying PD-1-interacting proteins was valid. We have provided more detailed descriptions in the main text, figure legend, and method section.

To further examine the list of potential PD-1 interacting proteins, we found that four deubiquitinases including USP5, USP10, USP15, and OTUB1, existed in the immuno-complex from cell lysates expressing PD-1 (**Revised Fig. 1a and Supplementary Table 1**). However, USP15, but not the other three deubiquitinases, was also identified in the immuno-complex from cell lysates expressing EV, suggesting that USP15 might unspecifically bind with anti-HA-agarose (**Supplementary Table 1**). Thus, USP15 should be a false positive for PD-1 interacting protein, which is consistent with our result that USP15 does not regulate PD-1 stability. Furthermore, although four deubiquitinases (USP5, USP10, USP15, and OTUB1) were identified as the potential PD-1 interacting proteins by MS analysis, the co-IP assay showed that USP5 is the most potent one to interact with PD-1 in cells, which might be the reason that USP5, but not other deubiquitinases, can regulate the PD-1 stability (**Revised Supplementary Fig. 2a**).

In fact, a recent study also identified the UPS5, USP10, and USP15 as the potential PD-1 interacting proteins by the MS analysis (**Zhou XA, et al. Proc Natl Acad Sci USA. 2020 Nov 10;117(45):28239-28250**). However, they did not further study and confirm which one is a *bona fide* deubiquitinase for regulating PD-1. As the sensitivity of MS assay is very high, it is possible that some identified proteins by MS assay might be false positive. Overall, we wish the reviewer can agree with us that we have provided more detailed information for the validity of the screen and enough biochemical evidence to support that USP5 is the most potent deubiquitinase for regulating PD-1 stability in this study.

Panel b: We repeated this experiment and obtained similar results that ectopic expression of USP5 WT, but not the enzymatically inactive USP5 (USP5-C335A) mutant, stabilized PD-1 at the protein level (**Revised Fig. 1b**). To determine the expression levels of myc-USP5, we used the anti-USP5 and anti-myc antibodies to detect the protein levels of endogenous and ectopic USP5 expression, respectively. The anti-USP5 antibodies can recognize both endogenous USP5 (lower band) and ectopic expression myc-USP5 (upper band). From the result, we can see that the ectopic expression levels of myc-USP5 was slightly higher than endogenous USP5, but their expression levels were comparable (**Revised Fig. 1b**).

Panel d: Great suggestions. As kindly suggested, we have drawn the graphic illustrations to show the shRNA-targeting sequence and its location on each of *USP5*, *USP10*, or *USP15* genes, respectively (**Revised Supplementary Fig. 1f**). To confirm that shRNAs targeting the *USP5*, but not *USP10* nor *USP15*, specifically reduced PD-1 expression, we simultaneously generated shUSP5, shUSP10, shUSP15, or shFGP stable MOLT-4 cell lines. We run these samples in one SDS-PAGE gel and blotted them in the same PVDF membrane. Although the knockdown efficiency of shUSP5, shUSP10, and shUSP15 is comparable, only knockdown of *USP5*, but not *USP10* nor *USP15*, could markedly decrease PD-1 expression in MOLT-4 cells (**Revised Fig. 1d**). These results further support our major conclusion that USP5, but not other deubiquitinases we examined, functions as a *bona fide* deubiquitinase for PD-1 in cells.

Panel J: As instructed, we have isolated the primary CD3⁺ T cells from C57BL/6J mice. After stimulation with anti-CD3/CD28 antibodies, we used various deubiquitinase inhibitors to treat the primary CD3⁺ T cells. The results showed that the EQAI3402143 is the most potent one to reduce PD-1 expression in the primary CD3⁺ T cells (**Revised Fig. 1j**), which is consistent with the results in Jurkat or MOLT-4 cells (**Revised Fig. 1i and Supplementary Fig. 1p**).

Moreover, we also purified the endogenous Usp5 protein using anti-USP5 antibodies from the mouse primary CD3⁺ T cells after stimulation with anti-CD3/CD28 antibodies and ubiquitinated PD-1 from HEK293T cells transfected with PD-1-cHA and His-ubiquitin. Subsequently, we performed the *in vitro* deubiquitination assay through co-incubating the purified Usp5 protein with ubiquitinated PD-1 with/without the EQAI3402143 treatment. The results demonstrated that purified USP5 dramatically removed the ubiquitination on PD-1, while the EQAI3402143 could inhibit USP5-mediated deubiquitination on PD-1 (**Revised Supplementary Fig. 2j**), providing evidence that the enzymatic activity of USP5 on removing PD-1 ubiquitination is directly inhibited *in vitro*.

Panel q: As kindly suggested, we have included the sample numbers in the revised figure legend (n = 5). We also reanalyzed the results and provided the following more information in the revised figure legend: Representative multiplex immunohistochemistry (mIHC) images of USP5 (red), PD-1 (green), CD3 (cyan) and DAPI nuclear staining (blue) in colon tumor section. White arrows indicate positive cells for PD-1 and USP5 colocalization on CD3⁺ T cells. Yellow color is considered overlapping for PD-1 and USP5 staining. Scale bars, 50 μm.

3) Figure 2: Panel A: Jurkat cells are abnormal lacking PTEN and other key enzymes. It is confusing since Jurkat cells generally express little PD1. Are the authors transfecting with PD-1 in these cells? The co-IP is convincing but needs to be convincingly demonstrated in activated primary mouse and human T-cells expressing PD1. Panels c-e: deletion analysis is fine as a first step but large deletions can have indirect effects due to conformation changes or changes in interactions with the membrane or associated proteins. It would strengthen the paper greatly if the authors could identify some specific residues (within motifs) needed for the interaction and to use these single or double mutated proteins to demonstrate effects. Panel h-I: convincing but confusing.

PD1 has been worked on for over a decade and it generally migrates as a single band on a gel with little evidence of the smear indicative of ubiquitination in primary cells- is this not the case? It may be that the overexpression of protein is somehow promoting the need for its degradation in a manner not normally seen in primary T-cells? A clear demonstration of PD1 Ub in normal T-cells is needed combined with a clear demonstration of the effects of USP5 in the KO primary T-cells.

Response: We thank the reviewer for raising these important concerns and providing us with great suggestions. As suggested, we have performed the following experiments to address the concerns raised for Figure 2:

Panel a: We are sorry for missing the information that Jurkat cells were treated with phytohemagglutinin A (PHA) to promote PD-1 protein expression before performing the co-IP assay. We have included the relevant information in the revised figure legends. As kindly instructed, we performed the co-IP in mouse primary CD3⁺ T cells with activation by using the anti-CD3/CD28 antibodies. Our results showed that Usp5 specifically existed in the immunoprecipitated complex of anti-PD-1, but not the control IgG, suggesting that Usp5 interacted with PD-1 in activated mouse primary T cells (**Revised Fig. 2b**).

Panel c-e: Great suggestions. Our results showed that both UBA1 and UBA2 domains are required for binding with PD-1. Through aligning protein sequences and comparing crystal structures of UBA1 and UBA2 domains in USP5, we found that the protein sequences and structures are very conserved (**Revised Supplementary Fig. 2d, e**). Several conserved amino acid residues located on the surface of crystal structure of UBA1 and UBA2 domains were identified, which might be potential key amino acid residues to mediate interaction with PD-1 in cells. These conserved amino acid residues were divided into four groups based on their positions on the crystal structure of UBA1 and UBA2 domains (**Revised Supplementary Fig. 2e**). To identify specific residues on UBA1/2 domains for PD-1 interaction, we mutated these conserved amino acid residues in each group into alanine residues on the UBA1/2 domains of USP5, designating the mutations as M1, M2, M3, and M4, respectively (**Revised Supplementary Fig. 2e**). We performed the streptavidin pull-down assays by incubating biotin-labeled synthetic pT234-PD-1 peptides with Flag-USP5 WT or different mutants. Compared with USP5 WT and other mutants, the interaction between M4 and pT234-PD-1 peptide was obviously reduced, suggesting the specific amino acid residues in M4 on the UBA1/2 domains of USP5 are critical for interaction with PD-1 (**Revised Supplementary Fig. 2f**).

Panel h-i: In the original Figure 2h-i (**Revised Fig. 2i-m**), we performed the *in vivo* ubiquitination assays to detect whether USP5 could remove the ubiquitination on PD-1. We agree with the reviewer that PD-1 generally migrates as single band on a gel when it is detected in the whole cell lysates. However, in our *in vivo* ubiquitination assays (see detailed description in the method section), total ubiquitinated proteins were firstly enriched by using the nickel-nitrilotriacetic acid (Ni-NTA) beads to pulldown His-tagged ubiquitinated proteins in the denaturing buffer (6 M guanidine-HCl, 0.1 M Na₂HPO₄/NaH₂PO₄, and 10 mM imidazole [pH 8.0]) and samples were subsequently detected with anti-PD-1 antibodies to determine whether PD-1 could be modified with ubiquitination, evidenced by the smear bands with high molecular weight. Moreover, recent studies showed that PD-1 can be ubiquitinated in the overexpression system of 293T cells,

activated human PBMCs, and Jurkat cells by using the similar *in vivo* ubiquitination assays (Meng X, et al. *Nature*. 2018, 564(7734):130-135; Lyle C, et al. *Sci Rep*. 2019, 9(1):20257; Zhou X, et al. *Proc Natl Acad Sci USA*. 2020, 117(45):28239-28250), which are consistent with our results.

As instructed, to further confirm whether PD-1 can be ubiquitinated in primary T cells, we isolated the *Usp5* WT and KO mouse primary CD3⁺ T cells and activated them with anti-CD3/CD28 antibodies. Subsequently, we performed the immunoprecipitation assay using the anti-PD-1 antibodies to enrich total PD-1 proteins including the form of ubiquitinated PD-1 in the denatured condition and utilized the anti-ubiquitin antibodies to detect the specific form of ubiquitinated PD-1. Our results demonstrated that the level of ubiquitinated PD-1 was increased in *Usp5* KO T cells compared to WT CD3⁺ T cells (Revised Fig. 2n). We have included these newly obtained results in the revised manuscript.

4) Figure 3: Panel a: PD1 flag is phosphorylated by several kinases in this assay system of over expression. ERK is just one of several kinases including AKT. The authors need to show specificity for ERK by titrating the kinase levels and to use sh/siRNA to KD ERK showing a loss of phosphorylation in primary T-cells. Does the single site phosphorylated when mutated, abrogate the effects of USP5 in modulating PD-1 function in primary T-cells? Is ERK more effective at lower levels of expression relative to the other kinases? Panel d is not adequate, numbers and sample size criteria are needed.

Response: We thank the reviewer for raising the concern and providing insightful suggestions. The following is our response to each concern raised in Figure 3.

Panel a: As instructed, we have repeated the experiment for gradiently increased expression of kinases and examined the effect of different expression levels of kinases on PD-1 protein abundance. Our results demonstrated that only ERK1, but not other kinases including AKT1, GSK3 β , and NEK9, dramatically increased PD-1 protein abundance, even ERK1 expression at a lower level relative to the other kinases (Revised Supplementary Fig. 3a), further supporting the notion that ERK might be the major kinase for regulating PD-1 phosphorylation and stability.

As suggested, we isolated primary CD3⁺ T cells from C57BL/6J mice and stimulated these cells using anti-CD3/CD28 antibodies. Subsequently, we used the shRNA to knockdown *ERK1/2* expression and detected the level of PD-1 phosphorylation at T234 site. The results showed that reduction of *ERK1/2* expression by shRNA decreased the level of PD-1 phosphorylation at T234 site detecting with the anti-pT234-PD-1 antibody (Revised Supplementary Fig. 4j). These newly obtained results suggest that knockdown of *ERK1/2* expression by shRNA reduces the PD-1 phosphorylation at T234 in primary T cells.

To address whether the mutation of a single phosphorylated site on PD-1 abrogates the effects of USP5 in modulating PD-1 stability in primary T-cells, we introduced the human PD-1 WT and T234A mutant into the mouse primary CD3⁺ T cells using the retro-virus expression

system. Subsequently, we treated these resulting cells with the USP5 inhibitor, EOAI3402143, as well as DMSO as a negative control. We detected the change of PD-1 WT and T234A proteins on the cellular surface of mouse primary CD3⁺ T cells using the human PD-1 antibody that specifically recognizes human, but not mouse PD-1. Our results showed that increased concentration of EOAI3402143 treatment gradually decreased the protein level of PD-1 WT, but not the T234A mutant, in the mouse primary CD3⁺ T cells, suggesting that the single phosphorylation site mutation on PD-1 could abolish USP5-mediated regulation of PD-1 stability (**Revised Supplementary Fig. 6f, g**). We have included these newly obtained results and discussion in the revised manuscript.

Panel d: Representative multiplex immunohistochemistry (mIHC) images of PD-1 (red), phosphor-ERK at Thr202/Try204 (p-ERK representing ERK activation, Green), CD3 (cyan) and DAPI nuclear staining (blue) in colon tumor sections. White arrows indicate positive cells for PD-1 and p-ERK colocalization in the CD3⁺ T cells. Yellow color is considered overlapping for PD-1 and p-ERK staining. n = 5. Scale bars, 50 μ m.

5) The authors also need to numbers of photo cuts of gels to allow the reader to see whether other bands exist or not.

Response: We agree with the reviewer that it will be great to provide the full uncut gels or blots to allow the future readers easily to follow. As the space limitation in the main figures, we have provided all the original full uncut gels or blots in the file of source data.

6) Figure 5: the generation of the KO mice is good. They now need to use the primary T-cells from these mice to show PD1 ubiquitination, ERK driven phosphorylation and the preferential effects of the loss of USP5 on PD-1 function and checkpoint blockade in these mice. Further in Panel e, although the effect is consistent with its importance; the authors need to include anti-PD1 treatment as a control- presumably anti-PD-1 may be more or less impactful due to altered PD1 expression? They also need to include several tumor models and some information on the nature of the altered expression of other receptors such as CD28, CTLA-4, TIM3 etc. as well as a nature and definition of memory vs effector memory and effector subsets. The authors should also consider crossing *pd1*^{-/-} mice with their *Usp5* KO mice where the phenotypes should be the same as the phenotypes seen in single KO mice?

Response: As instructed by the reviewer, we have isolated the primary CD3⁺ T cells from *Usp5* WT and cKO mice. After stimulation with anti-CD3/CD28 antibodies, *Usp5* WT and KO primary T-cells were lysed for the immunoprecipitation assay using the anti-PD-1 antibodies to enrich total PD-1 proteins including the form of ubiquitinated PD-1 in the denatured condition and utilized the anti-ubiquitin antibodies to detect the specific form of ubiquitinated PD-1. Our results

demonstrated that the level of ubiquitinated PD-1 was increased in *Usp5* KO CD3⁺ T cells compared to WT CD3⁺ T cells (**Revised Fig. 2n**). However, the phosphorylation of PD-1 at T234 was not dramatically changed between WT and *Usp5* KO CD3⁺ T cells (**Revised Supplementary Fig. 4k**). These results suggest that *Usp5* deficiency enhances PD-1 ubiquitination, but does not affect PD-1 phosphorylation at T234 site in T cells.

Previous studies have demonstrated that PD-1 plays an important function in regulating the activation and effector cytokine production of T cells (**Nishimura H, et al. *Immunity*. 1999, 11(2):141-51; Liu J, et al. *Proc Natl Acad Sci USA*. 2015, 112(21):6682-6687**). To explore the effects of the loss of *Usp5* on T cells, we first analyzed the T cell populations from the thymus or spleens of WT or *Usp5* cKO mice. Our results demonstrated that the percentages of CD4⁻CD8⁻ double-negative (DN), CD4⁺CD8⁺ double-positive (DP), CD8⁺ single-positive (CD8SP), and CD4⁺ single-positive (CD4SP) cells of total thymocytes are not significantly different between WT and *Usp5* cKO mice (**Revised Supplementary Fig. 7g, h**). Moreover, the percentages of naive (CD44^{low}CD62L^{high}), effector/effector memory (CD44^{high}CD62L^{low}, effector/EM), and central memory (CD44^{high}CD62L^{high}, CM) cells of total peripheral T cells isolated from spleens are also not significantly altered between WT and *Usp5* cKO mice (**Revised Supplementary Fig. 7i, j**). These results suggest that *Usp5* deficiency does not affect T cell development, the population of memory T cells at basal levels.

Furthermore, these isolated CD4⁺ and CD8⁺ T cells from the spleens of WT or *Usp5* cKO mice were stimulated with anti-CD3/CD28 antibodies *in vitro* and the T cell activation markers CD25 and CD69 were detected by the flow cytometry. The newly obtained results showed that the expression levels of CD69 and CD25 were significantly increased in *Usp5* cKO CD8⁺ T cells, but not in *Usp5* cKO CD4⁺ T cells, compared with WT cells upon stimulation with anti-CD3/CD28 antibodies (**Revised Supplementary Fig. 7k-n**). We examined the effector cytokines and cytotoxic molecule production of WT and *Usp5* cKO CD4⁺ or CD8⁺ T cells after treatment with anti-CD3/CD28 antibodies. The results demonstrated that ablation of *Usp5* in CD8⁺ T cells significantly enhances the production of cytokines (IFN γ and TNF) and cytotoxic molecule (GzmB) upon stimulation with anti-CD3/CD28 antibodies (**Revised Fig. 5c-e**). However, the production of IFN γ , TNF, and GzmB are not significantly different between WT and *Usp5* cKO CD4⁺ T cells after stimulation with anti-CD3/CD28 antibodies (**Revised Supplementary Fig. 7q-s**). These results together suggest that *Usp5* deficiency could promote the activation of CD8⁺ T cells *in vitro*.

To examine whether the *Usp5* expression affects Treg cell development, we utilized flow cytometry to analyze the Foxp3⁺/CD4⁺ Treg cells from the thymus or lymph node (LN) of WT or *Usp5* cKO mice. The newly obtained results demonstrated that there was no significant change of the Foxp3⁺/CD4⁺ Treg cells from the thymus or lymph node (LN) between WT and *Usp5* cKO mice (**Revised Supplementary Fig. 7k, l**), indicating that *Usp5* deficiency might not affect the development of Treg cells. To explore whether *Usp5* affects the suppressive functions of Treg cells, we co-cultured the effector CD8⁺ T cells (Teff) with WT or *Usp5* cKO Treg cells *in vitro* and detected the effect of Tregs on the proliferation of Teff using the proliferation marker Ki67. Compared to the group without Treg co-culture, the proliferation of Teff cells were significantly decreased when co-culturing with both WT and *Usp5* cKO Treg cells, suggesting the experimental

condition works well (**Revised Supplementary Fig. 7t**). However, there was no significant difference in the Ki67/CD8⁺ T cells co-culturing with WT compared to those co-culturing with *Usp5* cKO Treg cells (**Revised Supplementary Fig. 7t**). Moreover, the inhibitory cytokine production of IL10 and TFG- β in WT and *Usp5* cKO Treg cells was also not significantly changed (**Revised Supplementary Fig. 7u, v**). Together, these results demonstrate that *Usp5* deficiency might not affect the development and suppressive functions of Treg cells.

Of note, our results showed that *Usp5* cKO CD8⁺ T cells, but not CD4⁺ nor Treg cells, were significantly increased in the tumor microenvironment of subcutaneous MC38 and LLC tumors (**Revised Fig. 5m-p, 5w-y, and Supplementary Fig. 8a-c**). The PD-1 expression level was significantly decreased in the tumor-infiltrating *Usp5* cKO CD8⁺ T cells compared to WT CD8⁺ T cells (**Revised Fig. 5k, 5l, 5t-v**). These results together support our conclusion that *Usp5* deficiency enhances anti-tumor immunity largely through decreasing PD-1 and activating CD8⁺ T cells.

As kindly suggested, we further evaluated how *Usp5* deficiency in T cells affects the anti-PD-1 therapy. To this end, we subcutaneously transplanted the same number of MC38 colorectal cancer cells into *Usp5* WT and cKO C57BL/6J mice. Mice bearing MC38 tumors were divided into the following 4 groups: 1) WT mice with control IgG treatment; 2) WT mice with anti-PD-1 treatment; 3) cKO mice with control IgG treatment; 4) cKO mice with anti-PD-1 treatment. Our results demonstrated that anti-PD-1 antibody treatment significantly suppressed the tumor growth in WT mice compared with the control IgG treatment (**Revised Supplementary Fig. 10f, g**). However, although the tumor growth was significantly retarded in *Usp5* cKO mice compared with WT mice with control IgG treatment, there was no significant difference in tumor growth in *Usp5* cKO mice between the control IgG and anti-PD-1 therapy group (**Revised Supplementary Fig. 10f, g**), indicating that *Usp5* deficiency reducing the PD-1 protein abundance might mimic PD-1 blockade. Thus, USP5 inhibition in combination with anti-PD-1 therapy did not have an additive role in suppressing tumor growth.

To further validate this finding, we treated immunocompetent BALB/c mice bearing syngeneic CT26 tumors with the USP5 inhibitor EOAI3402143, anti-PD-1, alone or combined therapy (**Revised Supplementary Fig. 10h**). The single-agent treatment using EOAI3402143 or anti-PD-1 antibody significantly suppressed the tumor growth and enhanced the overall survival compared with the group of the control treatment (**Revised Supplementary Fig. 10i, j**). However, the combination of EOAI3402143 and anti-PD-1 treatment did not have an additive role in suppressing tumor growth or elevating the overall survival compared with the single-agent treatment (**Revised Supplementary Fig. 10i, j**). Taken together, these newly obtained results suggest that the USP5 inhibition-mediated reduction of PD-1 might mimic the PD-1 blockade.

Previous studies have demonstrated that the combinational therapy of PD-1 and CTLA-4 blockade is more effective than monotherapy in preclinical mouse tumor models and clinical trials (**Curran MA, et al., *Proc Natl Acad Sci USA*. 2010, 107(9):4275-80; Wei SC, et al., *Proc Natl Acad Sci USA*. 2019, 116(45):22699-22709; Larkin J, et al., *N Engl J Med*. 2019, 381(16):1535-1546**). We hypothesized that USP5 inhibition might improve the efficacy of anti-CTLA-4 immunotherapy *in vivo*. To test this hypothesis, immunocompetent BALB/c mice bearing CT26

tumors were treated with the USP5 inhibitor EOAI3402143, the anti-CTLA-4 antibody, alone or in combination (**Revised Supplementary Fig. 10k**). We observed that the treatment with EOAI3402143 plus an anti-CTLA-4 antibody resulted in seven complete responses out of nine treated mice, whereas the anti-CTLA-4 antibody alone only resulted in two complete responses of the nine treated mice (**Revised Fig. 6l**). Notably, the combination of EOAI3402143 with anti-CTLA-4 therapy significantly improved the overall survival compared with monotherapy (**Revised Fig. 6m**). In keeping with these findings, the EOAI3402143 treatment alone or combination with anti-CTLA-4 significantly reduced the PD-1 protein abundance in tumor-infiltrating CD3⁺ T cells and enhanced the production of IFN γ , TNF, and GzmB in tumor-infiltrating CD8⁺ T cells compared with control treatment (**Revised Fig. 6n-q**). Together, these results suggest that USP5 inhibition significantly enhances the efficacy of anti-CTLA-4 immunotherapy *in vivo*, which might be largely due to decreasing PD-1 and increasing cytotoxic activity of tumor-infiltrating CD8⁺ T cells.

As kindly instructed, we have included two other syngeneic mouse tumor models including mouse LLC lung cancer and EG7 lymphoma models in the revised manuscript. Consistent with subcutaneous MC38 tumors, both subcutaneous LLC and EG7 tumors grew slower in *Usp5* cKO mice than in WT mice (**Revised Fig. 5q-s and Supplementary Fig. 8h, i**). These results suggest the mechanism that *Usp5* deficiency in T cells enhances anti-tumor immunity and retards tumor growth might be general.

Moreover, the receptor PD-1, but not other receptors such as CD28, CTLA-4, and TIM3, was significantly decreased in infiltrating CD8⁺ T cells of LLC tumors derived from *Usp5* cKO mice compared with those from WT mice. Hence, these new results support our conclusion that *Usp5* deficiency enhances the anti-tumor immunity largely through reducing the PD-1 expression in CD8⁺ T cells.

We agree with the reviewer that it will be great to cross the *Pd-1*^{-/-} mice with *Usp5* cKO mice to further study whether the phenotypes of double KO mice are similar with the phenotypes observed in single KO mice. However, generating and obtaining results from the double KO mice will need at least one and a half year time and we have provided enough evidence from biochemical assays and several mouse tumor models to show that USP5 regulates anti-tumor immunity largely through controlling PD-1 protein abundance in CD8⁺ T cells and affecting the cytotoxic activity of CD8⁺ T cells in this updated manuscript. Thus, we wish the reviewer can agree with us that crossing *Pd-1*^{-/-} mice with *Usp5* cKO mice should be warranted in a separate study in the future, which has been discussed in the last paragraph of discussion section in the revised manuscript.

7) The description of the figures in the text is wholly inadequate. There is hardly any description of how experiments were done or text to guide the reader. One is just expected to accept a figure of results with a single line of description.

Response: We thank the reviewer for raising this concern and we are so sorry for missing detailed description of the figures in the text of the original submission. In the revised manuscript, we have added more description of the figures and how experiments were done in both the main text and the section of method.

Reviewer #3 (Remarks to the Author): with expertise in ubiquitination, immunology

Xiao et al identified USP5 as a PD-1-specific deubiquitinase using an IP-MS proteomic approach. They further discovered that Erk-mediated PD-1 phosphorylation facilitates USP5-PD-1 interaction and USP5-mediated PD-1 stabilization. Targeted deletion either in lymphoma cells by shRNA or conditional KO in mouse T cells, or pharmacological USP5 inhibition reduced PD-1 expression. As a consequence, inhibition of USP5 in T cells resulted in enhanced antitumor immunity. Overall, this is a novel study, and the data largely document their main conclusion, but with few concerns to be clarified.

Response: We sincerely thank the reviewer for recognizing the novelty of this study. We also thank the reviewer for acknowledging our efforts in providing enough results to support our major conclusion. At the same time, we also thank the reviewer very much for raising a few critical concerns and insightful suggestions, which have significantly improved our study. Following the reviewer's kind suggestions and instructions, we have obtained more results to experimentally address all the raised concerns as listed in the following point-to-point responses.

1. The authors generated the T cell specific USP5 conditional KO mice, but their phenotype is under characterized. Since PD-1 is expression in all T cell subsets including CD4, CD8 effector cells and Tregs, the impact of USP5 gene deletion to CD4 and CD8 T cell activation and Treg development and suppressive functions should be characterized. In particular, it is important to study whether USP5 impairs Treg suppressive functions to document their conclusion that the elevated antitumor immunity is due to the reduced CD8 PD-1 expression.

Response: We thank the reviewer for raising this concern and great suggestions. As instructed, we sorted the CD4⁺ and CD8⁺ T cells from the thymus or spleens of WT or *Usp5* cKO mice. Our results demonstrated that the percentages of CD4⁻CD8⁻ double-negative (DN), CD4⁺CD8⁺ double-positive (DP), CD8⁺ single-positive (CD8SP), and CD4⁺ single-positive (CD4SP) cells of total thymocytes are not significantly different between WT and *Usp5* cKO mice (**Revised Supplementary Fig. 7g, h**). Moreover, the percentages of naive (CD44^{low}CD62L^{high}), effector/effector memory (CD44^{high}CD62L^{low}, effector/EM), and central memory (CD44^{high}CD62L^{high}, CM) cells of total peripheral T cells isolated from spleens are also not significantly altered between WT and *Usp5* cKO mice (**Revised Supplementary Fig. 7i, j**). These results suggest that *Usp5* deficiency does not affect T cell development, the population of memory T cells at basal levels.

Furthermore, these isolated CD4⁺ and CD8⁺ T cells from spleens were stimulated with anti-CD3/CD28 antibodies *in vitro* and the T cell activation markers CD69 and CD25 were detected by the flow cytometry. The results showed that the expression levels of CD69 and CD25 were significantly increased in *Usp5* cKO CD8⁺ T cells, but not in the CD4⁺ T cells, compared with WT cells upon stimulation with anti-CD3/CD28 antibodies (**Revised Supplementary Fig. 7k-n**). Furthermore, we examined the effector cytokines and cytotoxic molecule production of WT and *Usp5* cKO CD4⁺ or CD8⁺ T cells after treatment with anti-CD3/CD28 antibodies. Our results demonstrated that ablation of *Usp5* in CD8⁺ T cells significantly enhances the production of

cytokines (IFN γ and TNF) and cytotoxic molecule (GzmB) upon stimulation with anti-CD3/CD28 antibodies (**Revised Fig. 5c-e**). However, the production of IFN γ , TNF, and GzmB are not significantly different between WT and *Usp5* cKO CD4⁺ T cells after stimulation with anti-CD3/CD28 antibodies (**Revised Supplementary Fig. 7q-s**). These results together suggest that *Usp5* deficiency could promote the activation of CD8⁺ T cells *in vitro*.

To examine whether the *Usp5* expression affects Treg cell development, we utilized flow cytometry to analyze the Foxp3⁺/CD4⁺ Treg cells from the thymus or lymph node (LN) of WT or *Usp5* cKO mice. The results demonstrated that there was no significant change of the Foxp3⁺/CD4⁺ Treg cells from the thymus or lymph node (LN) in WT and *Usp5* cKO mice (**Revised Supplementary Fig. 7k, l**), indicating that *Usp5* deficiency might not affect the development of Treg cells. To explore whether *Usp5* affects the suppressive functions of Treg cells, we co-cultured the effector CD8⁺ T cells (Teff) with WT or *Usp5* cKO Treg cells *in vitro* and detected the effect of Tregs on the proliferation of Teff using the proliferation marker Ki67. Compared to the group without Treg co-culture, the proliferation of Teff cells was significantly decreased when co-culturing with both WT and *Usp5* cKO Treg cells, suggesting the experimental condition works well (**Revised Supplementary Fig. 7t**). However, there was no significant difference in the Ki67/CD8⁺ T cells co-culturing with WT compared to those co-culturing with *Usp5* cKO Treg cells (**Revised Supplementary Fig. 7t**). Moreover, the inhibitory cytokine production of IL10 and TFG- β in WT and *Usp5* cKO Treg cells was also not significantly changed (**Revised Supplementary Fig. 7u, v**). Together, these results demonstrate that *Usp5* deficiency might not affect the development and suppressive functions of Treg cells.

Of note, our results showed that *Usp5* cKO CD8⁺ T cells, but not CD4⁺ nor Treg cells, were significantly increased in the tumor microenvironment of subcutaneous MC38 and LLC tumors (**Revised Fig. 5m-p, 5w-y, and Supplementary Fig. 8a-c**). The PD-1 expression level was significantly decreased in the tumor-infiltrating *Usp5* cKO CD8⁺ T cells compared to WT CD8⁺ T cells (**Revised Fig. 5k, 5i, 5t-v**). These results together support our conclusion that *Usp5* deficiency enhances anti-tumor immunity largely through decreasing PD-1 and activating CD8⁺ T cells. We have incorporated these newly obtained results and discussion in the revised manuscript.

2. Line up with point #1, the intratumoral CD4 T cells and Tregs should be characterized in Fig. 5 and 6.

Response: We thank the reviewer for providing us with this insightful suggestion. As suggested, we performed the multiplexed immunohistochemistry (mIHC) staining assay and fluorescence-activated cell sorting (FACS) assay to analyze the CD4⁺ T cells and Tregs in tumors. The results of mIHC showed that the tumor-infiltrating CD4⁺ T cells and Tregs were not significantly changed in MC38 tumors derived from WT and *Usp5* cKO mice (**Supplementary Fig. 8a-c**). The results of FACS analysis also demonstrated that there was no significant difference of intra-tumoral CD4⁺ T cells and Tregs derived from WT and *Usp5* cKO mice (**Revised Fig. 5m-p, 5w-y**). Moreover, there was no significant difference of the tumor-infiltrating CD4⁺ T cells and Treg cells in tumors

derived from WT and *Usp5* cKO mice with/without Trametinib treatment (**Revised Supplementary Fig. 10c-e**). These results further support our conclusion that *Usp5* deficiency enhances the anti-tumor immunity largely through decreasing PD-1 and enhancing the cytotoxicity of CD8⁺ T cells. We have incorporated these newly obtained results in the revised manuscript.

3. It will be interesting to test whether to simultaneously inhibit USP5-mediated PD-1 expression in cancer and CD8 T cells using PD-1+ cancer cells, such as EG7 lymphoma, further enhances antitumor immunity.

Response: Great suggestion. We first generated the mouse EG7 lymphoma stable cell lines with expressing shUsp5 as well as shGFP as a negative control. Our results showed that shUsp5 dramatically decreased the Usp5 protein expression, leading to a reduction of PD-1 protein in EG7 cells (**Revised Supplementary Fig. 8g**). Next, we subcutaneously grafted the shGFP and shUsp5 EG7 cells in WT and *Usp5* cKO C57BL/6 mice. Our results demonstrated that the growth of shUsp5-treated EG7 tumors was slower than those of shGFP-treated EG7 tumors in both WT and *Usp5* cKO C57BL/6 mice (**Revised Supplementary Fig. 8h, i**). However, the growth of shUsp5-treated EG7 tumors in *Usp5* cKO C57BL/6 mice was the most effectively retarded compared with those tumors derived from other groups (**Revised Supplementary Fig. 8h, i**). These newly obtained results provide the experimental evidence that simultaneously inhibiting the USP5-mediated PD-1 expression in cancer and CD8⁺ T cells might further enhance the antitumor immunity. We have included these results and related discussion in the revised manuscript.

4. The authors used a pan-Dub inhibitor EOAI3402143 that inhibits USP9X and USP24 in addition to USP5 for their studies, while the authors argue that USP9X and USP24 inhibition did not affect PD-1 expression, studies to determine whether this inhibitor alters other checkpoint receptors such as PD-L1 should be performed.

Response: We thank the reviewer for raising this concern. As kindly instructed, we have used the inhibitor EOAI3402143 to treat Jurkat cells and found that PD-1, but not other checkpoint receptors including PD-L1 and VISTA we examined, was dramatically reduced in Jurkat cells (**Revised Supplementary Fig. 1q**). These results suggest that the inhibitor EOAI3402143 might specifically regulate the PD-1 protein abundance via USP5 in cells.

5. Data In figure 3e, and figure 4d show that treatment with Trametinib inhibits USP5 expression and overexpression ERK increases USP5 expression as in figure 4a. The authors need to further

clarify whether the elevated USP5-PD-1 interaction is due to increased USP5 expression in addition to PD-1 phosphorylation.

Response: We thank the reviewer for pointing out the concerns. In the original Fig. 3e and Fig. 4d, we treated cells by using the increased concentration of Trametinib (1 μ M and 3 μ M). Only the high dosage of Trametinib (3 μ M) treatment slightly decreased the expression of USP5, whereas the low dosage of Trametinib (1 μ M) did not affect the protein expression level of USP5 in the WCL (**Original Fig. 3e and 4d**). However, the low dosage of Trametinib (1 μ M) could still reduce the USP5-PD-1 interaction, supporting the notion that USP5 interaction with PD-1 might be largely dependent on PD-1 phosphorylation (**Original Fig. 4d**). To further validate this notion, we repeated this experiment by using low dosages of Trametinib (0.5 and 1 μ M) to treat cells. The results showed that the low dosages of Trametinib treatment did not affect the USP5 expression, but still could disrupt the interaction of USP5 and PD-1 (**Revised Fig. 3e and 4d**).

In Figure 4a, we also repeated this experiment by reducing the transfection amount of HA-ERK1 plasmids and decreasing the expression of ERK1 kinase. The results demonstrated that the expression levels of ERK1 did not affect USP5 expression in the WCL (**Revised Fig. 4a**). However, the low levels of ERK1 kinase still increased the USP5-PD-1 interaction (**Revised 4a**). These results together suggest that ERK affects the USP5-PD-1 interaction might be largely through controlling PD-1 phosphorylation. However, it is possible that high dosage of Trametinib treatment or ERK kinase expression might also affect USP5 expression levels and its interaction with PD-1 in cells.

6. In figure 4e as well as in 3q, PD-1 appears to be increased in WCL when treated with Erk inhibitor, which inhibited PD-1-USP5 interaction. The authors need to validate this data or explain the discrepancy.

Response: We thank the reviewer for raising this concern and providing us with kind suggestions. In the original Fig. 3q and Fig. 4e, MOLT-4 and Jurkat cells were treated not only with ERK inhibitor, but also with the proteasome inhibitor MG132 to rescue the ERK inhibition-mediated PD-1 protein degradation. Thus, the PD-1 protein levels in WCL should be similar or a little bit less in the last two samples with both Trametinib and MG132 treatment compared with the first sample with only MG132 treatment. To validate these results and as kindly instructed, we repeated these experiments and changed the MG132 treatment time to make the PD-1 protein expression levels similar in the WCL. Our newly obtained results also showed that Trametinib treatment dramatically decreased the phosphorylation of PD-1 at T234 and the interaction between PD-1 and USP5 albeit the PD-1 protein levels in the WCL and IP are similar among three samples (**Revised Fig. 3q and 4e**).

7. The changes in PD-1 MFI on the surface of CD8 T cells in figure 6e and 6h are better to be provided.

Response: We thank the reviewer for the insightful suggestion. We fully agree with the reviewer that it will be better for providing the changes in PD-1 MFI on the surface of CD8⁺ T cells. For the original Fig. 6h, the results were analyzed by the flow cytometry and we reanalyzed the results to provide the PD-1 MFI changes on the surface of CD8⁺ T cells. The results showed that the PD-1 MFI was also significantly decreased on the surface of tumor-infiltrating CD8⁺ T cells derived from *Usp5* cKO mice compared with those from WT mice (**Revised Supplementary Fig. 10a**). However, for the original Fig. 6e, we did not provide the changes in PD-1 MFI on the surface of CD8⁺ T cells because the results for PD-1⁺ cells were qualified for the IHC staining results in the original Fig. 6D.

8. Several studies have detected that USP5 largely localize in nuclear, but data in Fig 1q show the exclusive USP5 cytoplasmic distribution, which should be validated in a better resolution and discussed accordingly.

Response: We thank the reviewer for raising this concern. Previous studies demonstrated that although USP5 has a cytoplasmic distribution, its major localization is in the nucleus, where USP5 is involved in regulating the DNA damage repair or heat-induced stress granules in cancer cells (Nakajima S, et al. *PLoS One*. 2014, 9(1): e84899; Xie X, et al. *J Cell Sci*. 2018, 131(8): jcs210856). However, a more recent study showed that USP5 largely localizes in the cytoplasm, whether USP5 interacts with NLRP3 to govern inflammasome activation (Nakajima S, et al. *PLoS One*. 2014, 9(1): e84899; Cai B, et al. *Autophagy*. 2022, 18(5):990-1004). As instructed, we have provided a better resolution picture for the original Fig. 1q and the result also showed that USP5 largely localizes in the cytoplasm of tumor-infiltrating CD3⁺ T cells albeit some signal for USP5 can be observed in the nuclear (**Revised Fig. 1q**). Together, these studies suggest that the predominant localization of USP5 in the cytoplasm or nucleus might be related to the cell types or different pathological conditions. We have included the discussion in the revised manuscript.

REVIEWERS' COMMENTS

Reviewer #1 (Remarks to the Author):

The authors have addressed my comments very well.

Reviewer #2 (Remarks to the Author):

The authors have addressed all my comments in a detailed and thorough manner. The paper is much improved.

Reviewer #3 (Remarks to the Author):

All my concerns have been addressed, there is no additional concerns. Overall this is a novel and important study. Acceptance for publication is recommended by this reviewer.

Point-by-point response to reviewers' comments

(NCOMMS-22-35135A)

First of all, we sincerely appreciate the three reviewers spending time on handling and reviewing our revised manuscript.

Reviewer #1 (Remarks to the Author):

The authors have addressed my comments very well.

Response: We sincerely thank the reviewer for recognizing our efforts in fully addressing the reviewer's concerns.

Reviewer #2 (Remarks to the Author):

The authors have addressed all my comments in a detailed and thorough manner. The paper is much improved.

Response: We sincerely thank the reviewer for acknowledging that we have thoroughly and experimentally addressed all the concerns, and the revised manuscript has been significantly improved.

Reviewer #3 (Remarks to the Author):

All my concerns have been addressed, there is no additional concerns. Overall this is a novel and important study. Acceptance for publication is recommended by this reviewer.

Response: We sincerely thank the reviewer for recognizing the novelty and importance of our study as well as our efforts in fully addressing all the concerns.